# How size and trigger matter: analyzing rainfall- and earthquake-triggered landslide inventories and their causal relation in the Koshi River basin, Central Himalaya

Jianqiang Zhang [1,2], Cees J. van Westen [2], Hakan Tanyas [2], Olga Mavrouli [2], Yonggang Ge [1], Samjwal Bajrachary [3], Deo Raj Gurung [3], Megh Raj Dhital [4], Narendral Raj Khanal[5]

[1]Key Laboratory of Mountain Hazards and Surface Process/Institute of Mountain Hazards and Environment, Chinese Academy of Sciences, Chengdu, China.

[2]Faculty of Geo-Information Science and Earth Observation (ITC), University of Twente, the Netherlands.

[3]International Centre for Integrated Mountain Development (ICIMOD), Lalitpur, Nepal.

[4]The Department of Geology, Tri-Chandra Multiple Campus, Ghantaghar, Kathmandu, Nepal.

[5]Central Department of Geography, Tribhuvan University, Kathmandu, Nepal.

*Correspondence to*: Jianqiang Zhang(zhangjq@imde.ac.cn)

**Abstract:** Inventories of landslides caused by different triggering mechanisms, such as earthquakes,extreme rainfall events or anthropogenic activities, may show different characteristics in terms of distribution, contributing factors and frequency-area relationships. The aim of this research is to study such differences in landslide inventories, and the effect they have on landslide susceptibility assessment. The study area is the watershed of the trans-boundary Koshi River in central Himalaya, shared by China, Nepal and India. Detailed landslide inventories were generated based on visual interpretation of remote sensing images and field investigation for different time periods and triggering mechanisms. Maps and images from the period 1992 to 2015 were used to map 5,858 rainfall-triggered landslides and after the 2015 Gorkha earthquake, an additional 14,127 co-seismic landslides were mapped. A set of topographic, geological and land cover factors were employed to analyze their correlation with different types and sizes of landslides. The frequency-area distributions of rainfall- and earthquake–triggered landslides have similar cutoff value and power-law exponent, although the ETL might have a larger frequency of smaller one. Also topographic factors varied considerably for the two triggering events, with both altitude and slope angle showing significantly different patterns for rainfall-triggered and earthquake-triggered landslides. Landslides were classified into two size groups, in combination with the main triggering mechanism (rainfall- or earthquake-triggered). Susceptibility maps for different combinations of landslide size and triggering mechanism were generated using logistic regression analysis. The different triggers and sizes of landslide data were used to validate the models. The results showed that susceptible areas for small and large size rainfall- and earthquake-triggered landslides differed substantially.

**Key words:** landslides, rainfall-triggered, earthquake-triggered, frequency-area analysis, susceptibility assessment, Nepal

34

## 1.  Introduction

Landslides are one of the most harmful geological hazards causing substantial fatalities and loss of property worldwide, affecting settlements, agriculture, transportation infrastructure and engineering projects (Dilley et al. 2005; Petley, 2012; Zhang et al., 2015; Haque et al., 2016). Among the various characteristics that determine the potential damage of landslides, size plays an important role, as well as velocity, depth, impact pressure, or displacement which differs for the various mass movement types. Volume may be an even more important landslide characteristic than size, but this is difficult to measure as it requires specific geophysical or geotechnical methods that can be applied at a site investigation level, or the use of multi-temporal Digital Elevation Models (SafeLand, 2015; Martha et al., 2017a). Therefore, empirical relations between landslide area and volume are generally used (Hovius et al, 1997; Dai and Lee, 2001; Guzzetti et al., 2008; Larsen et al., 2011; Klar et al., 2011; Larsen and Montgomery, 2012). To investigate whether earthquake- and rainfall-triggered landslides inventories have similar area-frequency distributions, area-volume relations and spatially controlling factors, it is important to collect event-based landslide inventories. The difficulty is to collect complete inventories that are independent for earthquakes and rainfalls in same study area.

The quality of a landslide inventory can be indicated by its accuracy,  which refers to the correctness in location and classification of the landslides , and its completeness , which measures how many of the total number of landslides in the field were actually mapped (Guzzetti et al., 2012) . The accuracy and completeness have a large influence on the quality and reliability of the susceptibility and hazards maps that are either using the inventory as input (e.g. in statistical modelling) and in validation (e.g. statistical and physically-based modeling) (Li et al., 2014). There are several explanations why landslide inventories differ in  frequency-area distribution, such as the under sampling of small slides (Stark and Hovius, 2001), or the amalgamation, the merging of several landslides into single polygons (Marc and Hovius 2015).

Landslides might be triggered by various processes, among which anthropogenic activities, volcanic processes, sudden temperature changes, earthquakes and extreme rainfall (Highland and Bobrowski, 2008). The latter two are the most frequently occurring, and causing the highest number of casualties (Keefer, 2002; Petley, 2012; Kirschbaum et al, 2015; Froude and Petley, 2018). Comparing landslide inventories for the same area and for the same triggering event has been carried out by several authors (e.g. Pellicani and Spilotro, 2015; Tanyas et al., 2017a). Some studies took independent earthquake- and rainfall-triggered landslide inventories to compare the characteristics of landslides induced by different triggers. Malamud et al. (2004) compared earthquake-triggered landslides from the Northridge earthquake, Umbria snowmelt-triggered landslide and Guatemala rainfall-triggered landslide as examples, and concluded that the three frequency-area distributions were in good agreement with each other. Meunier et al. (2008) compared earthquake-triggered landslides, from Northridge, Chi-Chi Finisterre Mountains (Papua New Guinea), to

evaluate topographic site effects on the distribution of landslides. Tanyas et al. (2017b) created a database with 363 landslide–triggering earthquakes and 64 digital landslide inventories, which were compared. The number of studies that compare earthquake-triggered landslide with rainfall triggered ones for the same area is less numerous. They are mostly focusing on mapping rainfall-induced landslides after an earthquake, such as for the 1999 Chi-Chi earthquake (Lin et al., 2006; 2008), the 2005 Kashmir earthquake (Saba et al., 2010) or the 2008 Wenchuan earthquake (Tang et al., 2010; Tang et al., 2016; Fan et al., 2018a). Fewer studies carried out on multi-temporal RTL inventories in Taiwan, Papua New Guinea, Japan and Central Nepal before earthquake, which supplied good comparison study for RTL under the effect and without the effect of earthquakes (Marc et al., 2015, 2019). The problem with the studies indicated above is that rainfall-triggered landslides that occur shortly after a major earthquake are generally following the same spatial patterns, due to the availability of large volumes of landslide materials of the co-seismic landslides (Hovius et al., 2011; Tang et al., 2016; Fan et al., 2018a). However, other studies argue that there is not a clear correlation of rainfall-triggered landslides with the co-seismic pattern, as only the 20- 30% of the RTL that occurred just after an earthquake, are spatially related to ETL, suggesting limited re-activation of ETL by RTL (Marc et al., 2015, 2019).

There are very few studies that have validated landslide susceptibility maps with independent landslide inventories of triggering events that occurred after the maps were produced. Chang et al. (2007) used landslides triggered by a major earthquake and a typhoon prior to the earthquake to develop an earthquake-induced model and a typhoon-induced model. The models were then validated by using landslides triggered by three typhoons after the earthquake. According to the results, typhoon-triggered landslides tended to be near stream channels and earthquake-triggered landslides were more likely to be near ridge lines. Although landslide size is often considered important in hazard and risk assessment, it is generally not considered as a separate component of the susceptibility assessment. The different relation with contributing factors of earthquake-triggered and rainfall-triggered landslides may also be related to the size distribution (Korup et al., 2007). For instance, Fan et al. (2012) concluded that small ($<10\times10^4 m^3$) rainfall-triggered landslide and earthquake-triggered landslides have similar runout distances, whereas for larger landslides earthquake-triggered ones showed longer runouts. Peng et al. (2014) analyzed the landslides in the Three Gorges area and found that different landslide sizes had different relations with contributing factors.

The aim of this study is to investigate the differences in the characteristics of earthquake-triggered and rainfall triggered landslides in terms of their frequency-area relationships, spatial distributions and relation with contributing factors, and to evaluate whether separate susceptibility maps generated for specific landslide sizes and triggering mechanism are better than a generic landslide susceptibility assessment including all landslide sizes and triggers. This research aims to address a number of questions related to the difference of using earthquake-induced and rainfall-induced landslide inventories for the generation of landslide susceptibility maps. The question will be addressed that, whether different landslide size groups are controlled by different sets of contributing factors. By extension, whether it

is possible to utilize inventories of earthquake-triggered landslides (ETL) as inputs for analyzing the susceptibility of rainfall-triggered landslides (RTL) and vice versa.

## 2. Study area

The study was carried out in the Koshi River basin, which is a trans-boundary basin located in China, Nepal and India in the central Himalayas (Fig. 1a). The mountainous regions in the upper reaches of the basin where landslides have occurred are located in China and Nepal, and the Indian part consists of relatively flat areas. The elevation of Koshi River basin varies from 60 m a.s.l. at the outlet in India up to 8,844 m at the highest point at Mount Everest. The Koshi basin can be classified into 6 physiographic zones from South to North: Terai, Siwalik Hills, Mahabharat Lekh, Middle Mountains, High Himalaya,and Tibetan Plateau (Gurung and Khanal, 1987; Dhital, 2015). Considering the distribution of landslides, the Tibetan plateau in the upper reaches and the plains in the lower reaches were excluded.

In the Koshi Basin, the major geological structures have an approximate east–west orientation, such as the foreland thrust-fold belt, Main Central Thrust (MCT), South Tibetan Detachment System (STDS) and the Yarlung Zangbo Suture Zone (YZSZ) (Gansser, 1964; Dhital,2015). The southernmost part of the basin consists of the Quaternary sediments underlain by the Neogene Siwaliks. The Siwaliks comprise soft mudstones, sandstones and conglomerates. In this part of the foreland basin, a number of emergent and blind imbricate faults originate from the Main Himalayan Thrust. The overlying Lesser Himalayan succession forms duplexes and imbricate stacks. The Proterozoic to Miocene rocks of the Lesser Himalaya include limestones, dolomites, slates, phyllites, schists, quartzites, and gneisses (Dhital, 2015). A regional-scale thrust MCT separates the Lesser Himalayan sequence from the overlying Higher Himalayan crystallines, which consist of medium- to high-grade metamorphic rocks (e.g., schists, quartzites, amphibolites, marbles, gneisses, and migmatites) and granites aged from the Proterozoic to Miocene. The STDS delineates the Higher Himalayan rocks from the overlying Tethyan sedimentary sequence of Paleozoic–Cenozoic age (Gansser, 1964; Burg et al., 1984; Hodges et al., 1996) (Fig. 1b).

In the study area there are three main tributaries of the Koshi River: the Arun (main branch) coming from the north, the Sun Koshi from the west and Tamor from the east. Nearly every year, during the monsoon period, which generally lasts from June to September, the area is affected by rainfall-triggered landslides. Dahal and Hasegawa (2008) used a dataset of 193 landslides occurring from 1951 to 2006, part of which were from the Koshi River basin, to generate a threshold relationship between rainfall intensity, rainfall duration, and landslide initiation. The latest research from Marc et al.( 2019) gives the magnitude of annual landsliding in different High Himalayan valleys.

The area was severely affected by the Gorkha earthquake, with a moment magnitude of 7.8 on 25 April 2015. The epicenter was located near Gorkha, which is about 80km west of the study area. A second major earthquake occurred along the same fault on 12 May 2015 with a moment magnitude of 7.3 with the epicenter located inside the Koshi

River basin. The second event is considered as a major aftershock of the main Gorkha earthquake. Both events
triggered many landslides (Collins and Jobson, 2015; Kargel et al., 2016; Zhang et al., 2016; Martha et al., 2017b).

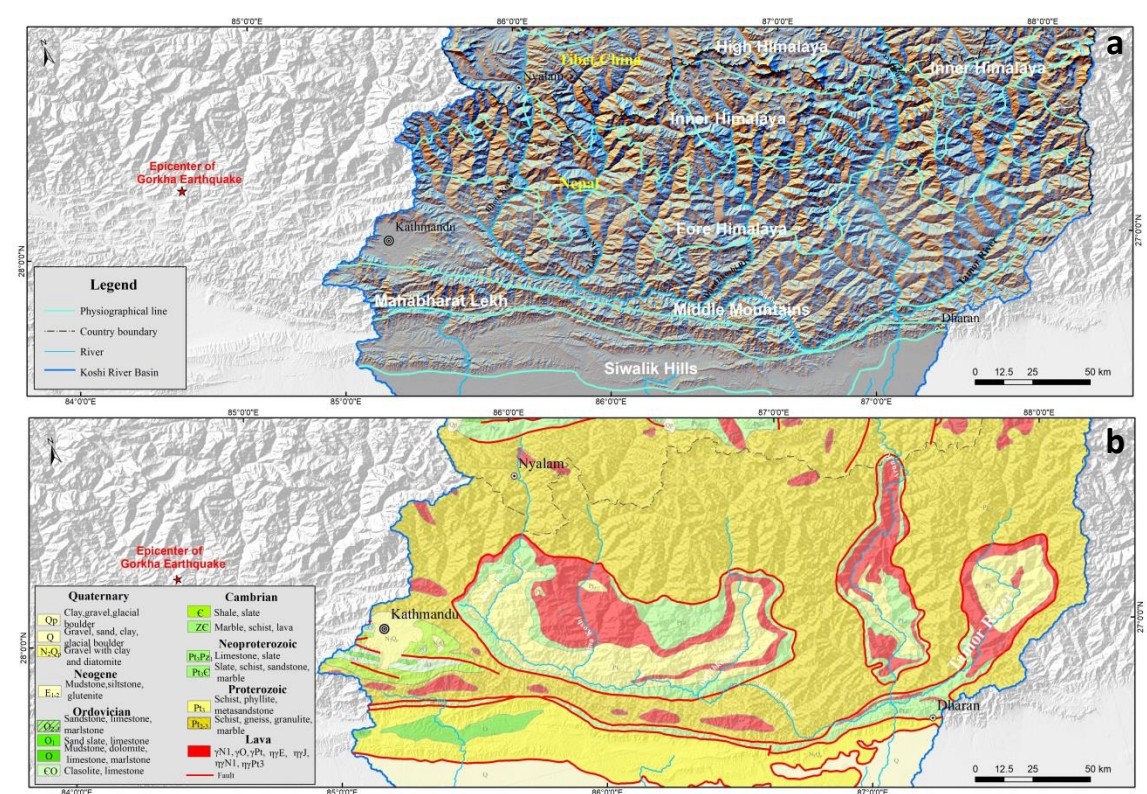

**Fig. 1** Maps showing the study area (a) Physiographic zones of the Koshi River basin; (b) Geological map showing the
main geological zones (Dhital, 2015; Zhang et al., 2016).

### 3. Input data

The study requires a series of landslide inventory maps, and contributing factor maps, which were generated for the
middle part of the Koshi basin, where most of the landslides were concentrated. Two landslide inventories were
generated: a pre-2015 inventory showing rainfall-triggered landslides, and a co-seismic landslide map for the 2015
Gorkha earthquake. The pre-2015 inventory map was generated using topographic maps, multi-temporal Google Earth
Pro images and Landsat ETM/TM images. We were able to digitize landslide polygons from the available 1:50,000
scale topographic maps, which cover only the Nepalese part of the Koshi River basin. These maps were generated
from aerial photographs acquired in 1992, and active landslides with a minimum size of 450 $m^2$ visible on these
images were marked as separate units. The landslides could not be separated in initiation and accumulation zones, and
also no classification of landslide types could be done, as this was not indicated on the topographic maps. A set of pre-
2015 Landsat ETM/TM images were available for the entire study area, from which the post 1992 and pre-2015
landslides. Pre-2015 landslides were also mapped from historical images using Google Earth Pro Historical Imagery
Viewer which contains images from 1984 onwards. Although the oldest images are Landsat images, the more recent
ones have much higher resolution, although not covering the whole study area in equal level of detail. By comparing
the different images for the period between 1992 and 2015 we were able to recognize most of the landslides. We
carried out field verification for a number of samples (Fig. 2). Images from Google Earth were downloaded and geo-
referenced and landslides were mapped using visual image interpretation and screen digitizing. A total of 5,858 rainfall
induced landslides were identified in the Koshi River basin. This inventory has a limitation that, landslide occurred and
revegetated during 1992 and 2015 could not be identified by the remote sensing images obtained in 2015. It is
impossible to make a complete historical landslide inventory in this region due to lack of multi-temporal high
resolution images (Marc et al., 2019).
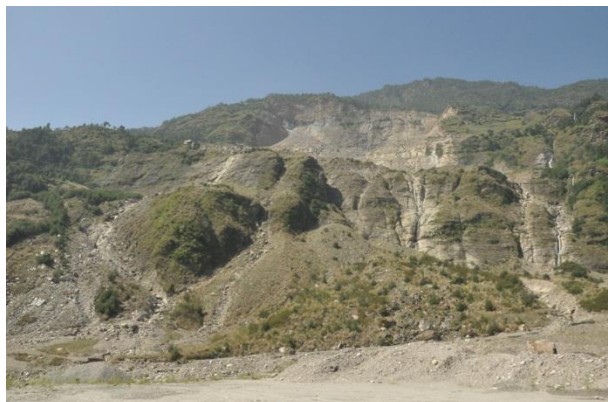 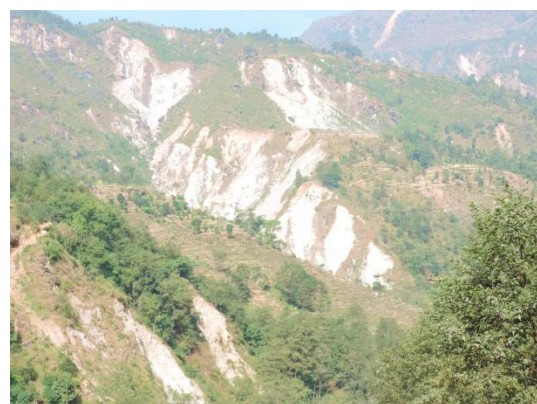
Fig. 2 Field investigation on landslide in Koshi river basin (a) Jury landslide triggered by rainfall occurred on August 2,
2014 at Sunkoshi river(photo by Bintao Liu in 2017); (b) Small size of landslides triggered by Gorkha earthquake in
Bhotekoshi watershed.

After the 2015 April 25[th] Gorkha earthquake, a substantially complete earthquake-triggered landslide inventory was
created by Roback et al. (2017). They mapped landslides using high-resolution (<1m pixel resolution) pre- and post-
event satellite imagery. In total 24,915 landslide areas were mapped, of which 14,022 landslides were located in the
Koshi river basin. Chinese GaoFen-1 and GaoFen-2 satellites imageries (with 2.5m resolution) of the CNSA (China
National Space Administration), which are part of the HDEOS (High-Definition Earth Observation Satellite) program,
were employed to validate this landslide inventory. These images were captured during 27 April, 2015 to May 14 2015.
Finally 15 landslide polygons were deleted, and 120 landslides were added to the inventory.
For the susceptibility assessment, we extracted the point located in the highest part of the landslides, as indicative of
the initiation conditions. Different DEMs, such as ASTER GDEM, SRTM Digital Elevation Model with both 90 m and
30m spatial resolution, as well as ALOS PALSAR DEM were evaluated to use in this study. After careful analysis
however, both ASTER GDEM and 30m SRTM contained many erroneous data points, ALOS PALSAR DEM with
highest resolution of 12.5m, was utilized in this study. ESRI ArcGIS software enabled the calculation of topographical
factors including slope gradient, aspect, and curvature. Streams and gullies were obtained through DEM processing,
and the drainage density was calculated. The land cover dataset GlobeLand30 with $30 \times 30$m spatial resolution,
developed by the National Geomatics Center of China, was employed in this study. The land cover types include
cultivated land, forest, grassland, shrub land, wetland, water bodies, tundra, artificial surfaces and bare land.
Geological maps of Nepal, and Tibet were obtained from Chengdu Geological Survey Center of the China Geological
Survey. The Peak Ground Acceleration data for the Gorkha earthquake were obtained from USGS Shakemap, which
was designed as a rapid response tool to portray the extent and variation of ground shaking throughout the affected
region immediately following significant earthquakes (Wald et al., 1999). Given the rather low resolution of the input
data, the relation with landslides as small as $50m^2$ may not be optimal, especially also considering the rather long time
period over which land cover changes have occurred in many areas. But given the regional scale of this analysis, the
use of higher resolution data was unfortunately not a viable option.

## 4. Methods

Figure 3 gives an overview of the method followed in this study. The landslide inventories were subdivided into
training and test datasets. It is a generally accepted method in literature to separate the landslide dataset into a training
and validation set (e.g. Hussin et al. 2016; Reichenbach et al., 2018), although the separation thresholds differs among
authors. We decided to select 60% of the landslide data as training data for the modeling, and 40% for the validation.
We examined the frequency-area distribution of the gathered inventories using the method described by Clauset et al.
(2009). They proposed a numerical method to identify the slope of power-law distribution (β) and the point where
frequency-area distribution diverges from the power-law (cutoff point).
Based on the frequency area distribution the RTL and ETL inventories were separated in two size-groups each. Initially
bivariate statistical analysis was used for the different types and sizes of landslides, to investigate the correlation
between landslides with contributing factors. After selecting the relevant factors, the logistic regression method was
used to build the susceptibility model for each size group. The Logistic Regression method is the most commonly used
model in landslide susceptibility assessment (Ayalew and Yamagish, 2005; Bai et al., 2010; Das et al., 2000; Nandi and
Shakoor, 2010; Wang et al., 2013). For the susceptibility modeling of RTL, the following factors were used: altitude
($x_1$), slope gradient ($x_2$), curvature ($x_3$), slope aspect ($x_4$), relative relief ($x_5$), drainage density ($x_6$), lithology ($x_7$),
distance to faults (x₈) ,land cover type (x₉), precipitation during monsoon(x₁₀). For the susceptibility modeling of ETL,
precipitation during monsoon($x_{10}$) was instead of peak ground acceleration ($x_{10}$). The statistical software R developed
at Bell Laboratories was used to build the models for different types and sizes of landslide respectively. ROC (Receiver
Operating Characteristic) curves (Fawcett, 2006) were used to verify the accuracy of the susceptibility models, and
finally six landslide susceptibility maps were generated and compared (Fig. 3).

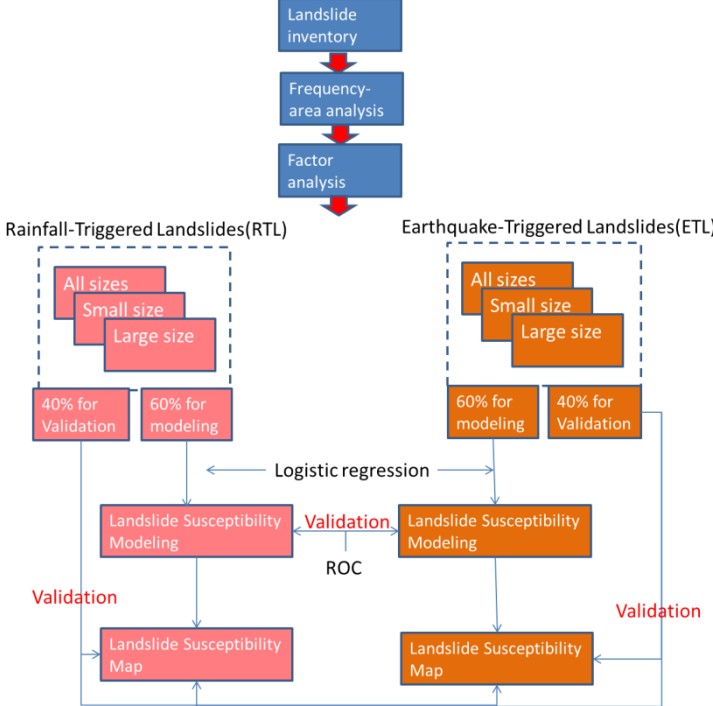


**Fig. 3** Methodology for susceptibility assessment to different types and sizes of landslide

## 5. Landslide characteristics
In the Koshi River basin, a total of 5,858 RTL were mapped. The Gorkha earthquake triggered more than 25,020
landslides, of which 14,127 were located in the Koshi River basin. Landslide characteristics were analyzed based on
frequency-area distribution and factor statistics (Fig. 4).

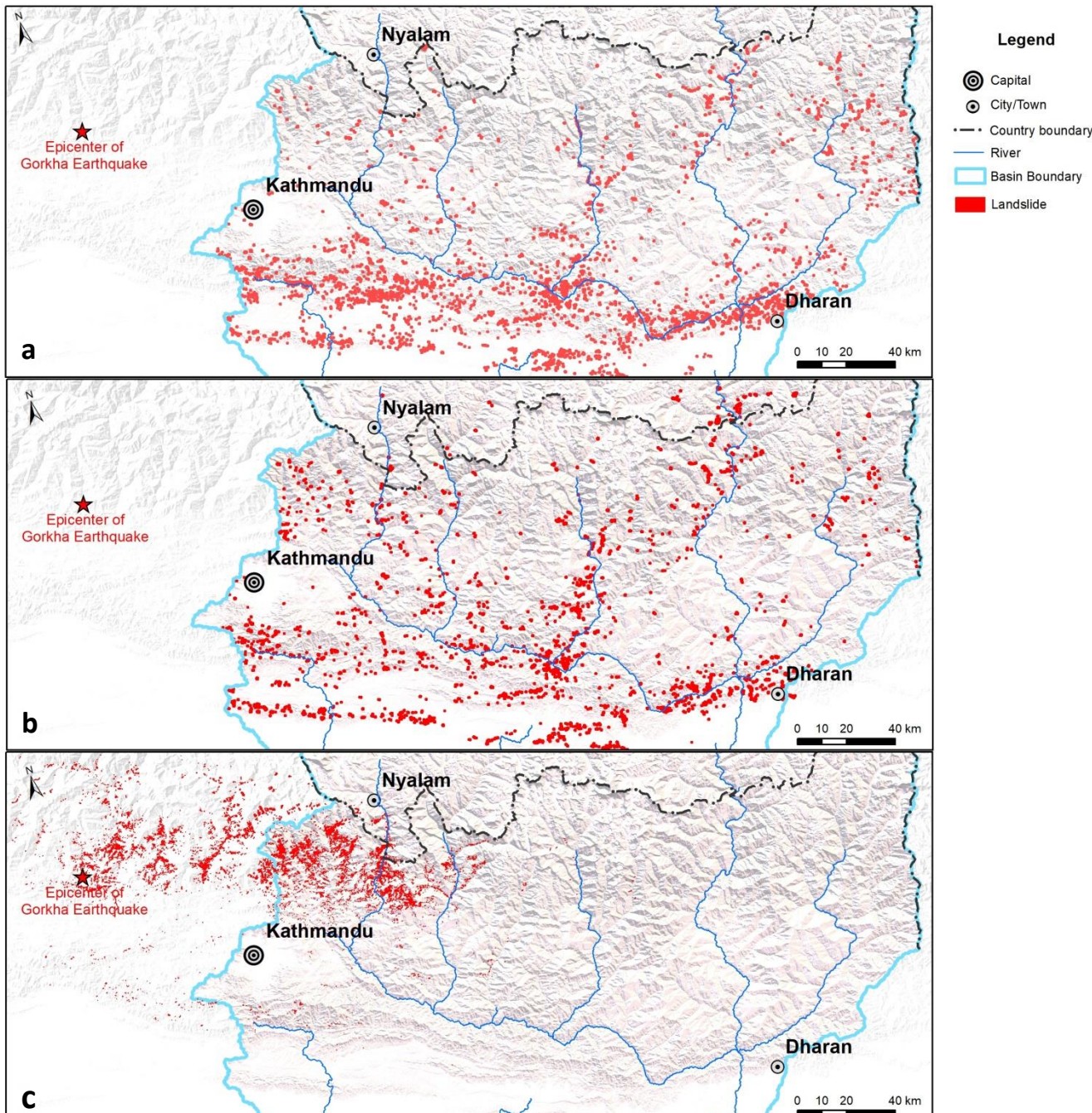


**Fig. 4** Landslide inventories of the Koshi River basin (a) Rainfall induced landslide inventory of events before 1992; (b)

Rainfall induced landslide inventory for the period between 1992 to 2015; (c) Inventory of landslides triggered by the

2015 Gorkha earthquake(Roback et al. 2017).

## 5.1 Landslide frequency-area distributions

Size statistics of landslides are analyzed using frequency-area distribution curves of landslides (e.g., Malamud et al., 2004). There is a large literature arguing that frequency-area distribution of medium and large landslides has power-law distribution, which diverges from power-law towards smaller sizes (e.g., Hovius et al., 1997, 2000; Malamud et al., 2004). Given this argument, we can identify the divergence point of frequency-area distribution curve to determine a site specific threshold values referring to the limit between medium and small landslides.

The frequency-area distributions (FAD) of landslides were separately analyzed for both RTL and ETL inventories (Fig. 5). For the RTL both landslide inventory datasets of before 1992 and 1992~2015 were analyzed (Fig. 5a). For the ETL of the Gorkha earthquake, landslides located in the Koshi River basin were analyzed separately from the entire landslide-affected area. We obtained similar $\beta$ values for the RTL triggered before 1992 ($\beta = 2.44$) and triggered from 1992 to 2015 ($\beta = 2.38$) (Fig. 5a). On the other hand, we observe larger differences between the $\beta$ values obtained for ETL inventories created for both Koshi River basin and entire landslide-affected area (Fig. 5b).

We also examine the cutoff values of inventories. The historical RTL inventories and ETL inventory that we examined for both Koshi River basin and entire landslide-affected area gave similar cutoff values changing from 24,884 $m^2$ to 32,913 $m^2$ (Fig. 5). This finding shows that, the limit between small and large landslides are consistently obtained from these inventories about 30,000 $m^2$. Given this finding, the proposed landslide size classification system of China the Tong et al. (2013) seems like an acceptable approach for our study area. They proposed a classification with landslides with an area smaller than 10,000 $m^2$ as small, those with an area between 10,000 $m^2$ and 100,000 $m^2$ as medium, and those with larger sizes than 100,000 $m^2$ as large size landslide. Considering this study, and the cutoff values calculated in our study, 30,000 $m^2$ was picked as a threshold value for large landslides.

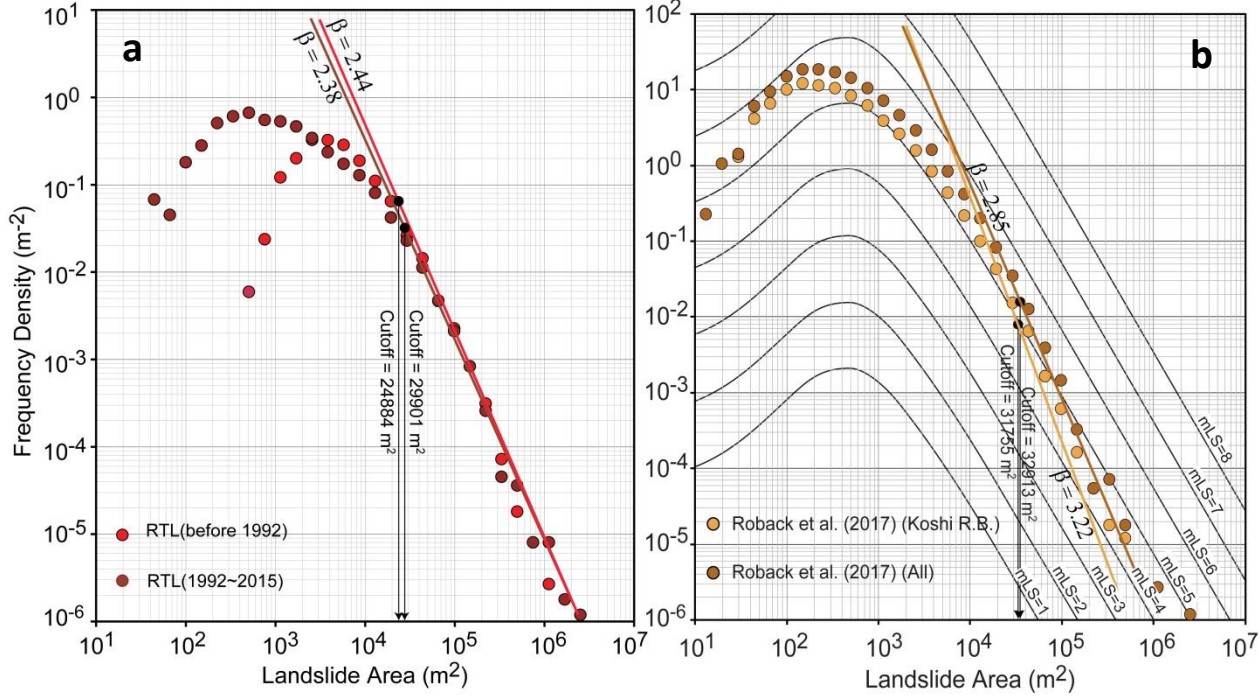

**Fig. 5** Landslide frequency - area distributions of (a) RTL inventories, (b) ETL inventories created for Koshi River basin and ETL inventories created for the entire landslide-affected area of the 2015 Gorkha, Nepal earthquake(Roback's landslide inventory was validated). Cutoff and β values are calculated using the method proposed by Clauset et al. (2009).

Based on the results of the FAD analysis, that resulted in similar cutoff values for the RTL and ETL and similar β values, we subdivided them into two size-groups, with 30,000 m² as threshold value (Table 1). The results will therefore be more reliable for the class above the threshold of 30,000 m², where under sampling is not an issue, then for the small landslide class, which has different rollover points, and completeness levels.

**Table 1** Numbers for different types and sizes of landslide in Koshi River basin

|  | Rainfall-triggered landslides (RTL) | | | Earthquake-triggered landslides (ETL) | | |
| --- | --- | --- | --- | --- | --- | --- |
|  | All sizes | Small size | Large size | All sizes | Small size | Large size |
| Total | 5,858 | 5267 | 591 | 14,127 | 13981 | 146 |
| Modelling | 3,515 | 3160 | 355 | 8476 | 8388 | 88 |
| Validation | 2,343 | 2107 | 236 | 5650 | 5593 | 58 |

253

## 5.2 Correlation of landslides with contributing factors

In order to evaluate their relation with landslide occurrence the factor maps were analyzed using the Frequency Ratio method (Razavizadeh et al., 2017).

$$FR = \frac{E/F}{M/L}$$

where $E$ is the area of landslides in the conditioning factor group, $F$ is the area of landslides in the entire study area, $M$ is the area of the conditioning factor group, and $L$ is the entire study area. The analysis was carried out for different triggers and size groups, and each time two factors were combined (e.g. altitude with slope gradient, altitude with slope direction, lithology with slope gradient). The results are summarized in Fig. 6. Fig. 6a&b show that rainfall triggered landslides (RTL) are more frequent in low altitude areas then earthquake triggered landslides (ETL). However, it is important to keep in mind that the ETL is an event inventory of a single earthquake, where the epicenter was located at higher altitude (See Fig. 4) and the RTL is a multi-temporal inventory, showing the accumulated inventory of many individual events.

Fig. 6 c&d show the relation with slope and lithology. RTLs are concentrated on Proterozoic metamorphic lithological units (Pt3), consisting of schist, phyllite and metasandstone, and in Quanternary molasse ( N2Qp ) units, consisting of gravel and clay (See Fig. 1). ETLs are linked to units consisting of shale and slate (Pt3ϵ), and Cambrian units consisting of shale and slate (ϵ) and marble, schist and lava (Zϵ).


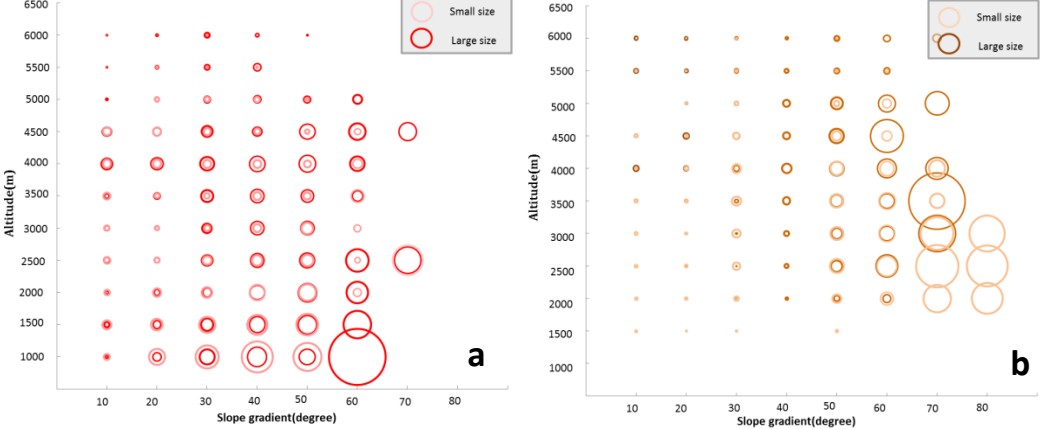


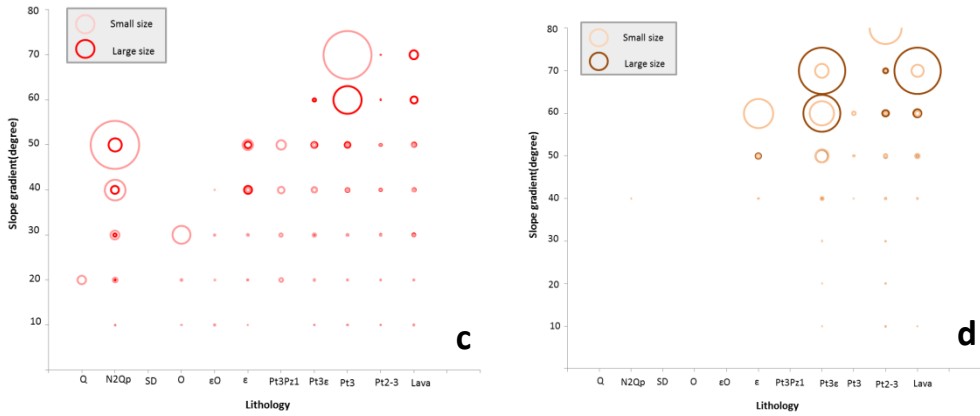


**Fig. 6** Correlation between landslides and other factors for rainfall triggered landslides (RTL) on the left side, and earthquake-triggered landslides (ETL) on the right side. The size of the circles indicate the value of the Frequency Ratio. a & b: Relation between altitude and slope gradient; c & d: Relation between Lithology and slope gradient.


## 6. Landslide susceptibility assessment

6.1 Landslide susceptibility models

The following factors were used for the susceptibility modeling of RTL: altitude($x_1$), slope gradient($x_2$), curvature($x_3$), slope aspect($x_4$), relative relief($x_5$), drainage density($x_6$), lithology($x_7$), distance to fault($x_8$),land cover type($x_9$) and precipitation during monsoon($x10$). Peak Ground Acceleration (PGA) was used instead of precipitation for the susceptibility modeling of ETL (Fig. 7). The R software was used to build the models by Logistic Regression method for different types and sizes of landslide respectively (Table 2). ROC curves were generated to verify the accuracy of each susceptibility model, and value of the Area Under Curve (AUC) was calculated (Table 2).

The coefficients for the contributing and triggering factors in the landslide susceptibility models show differences between triggers and different sizes of landslides. Curvature, altitude and slope gradient have a high impact on the susceptibility of RTL, while curvature, PGA, relative relief, and slope gradient have high impact on susceptibility of ETL. The size classes of RTL show larger differences in weight of curvature, relative relief and altitude. For ETL the difference between size classes are largest for factors of PGA, curvature, and relative relief.







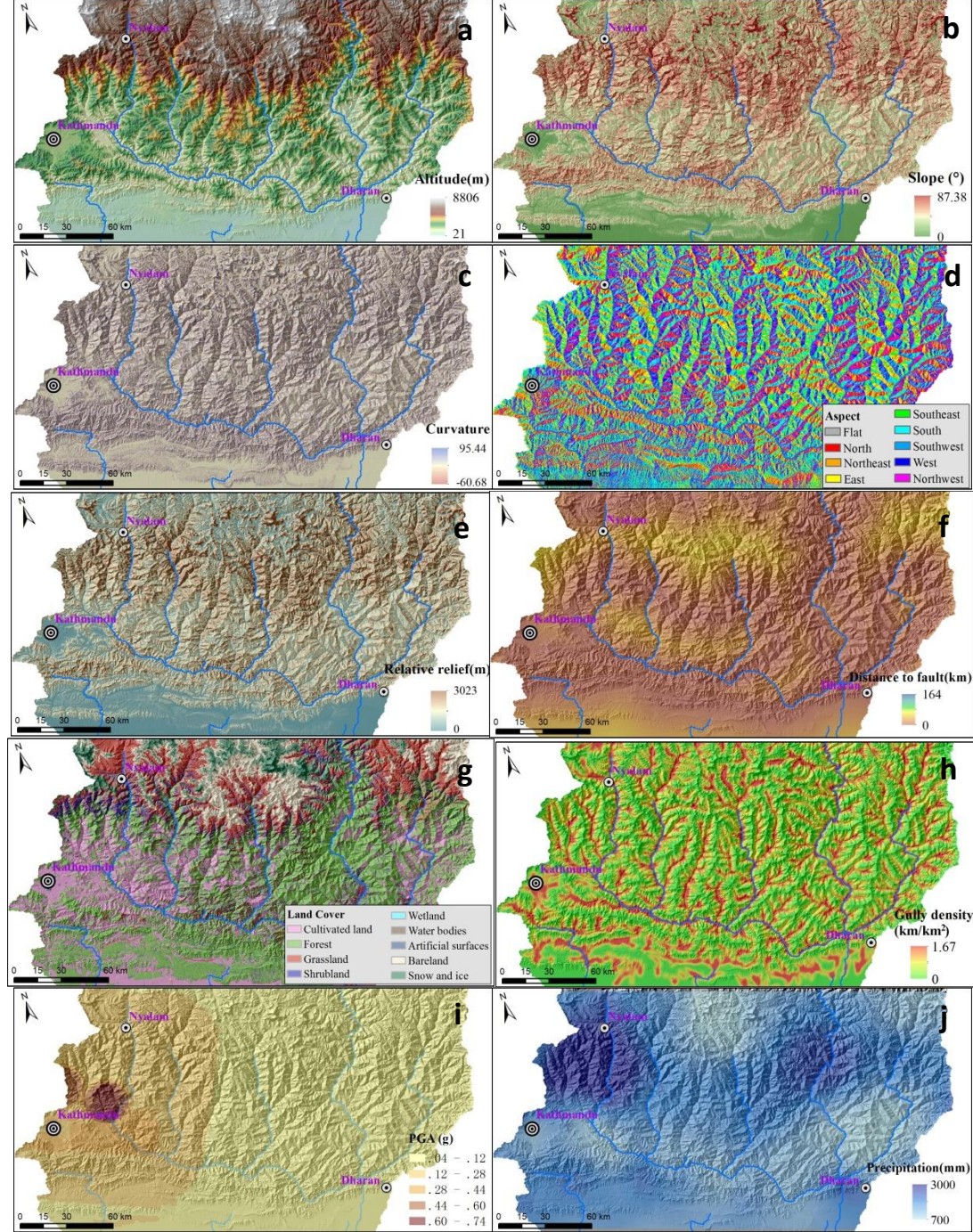

**Fig. 7** Landslide susceptibility assessing factors; a: altitude(Data source: JAXA/METI ALOS PALSAR DEM); b: slope

gradient; c: slope curvature; d: slope direction; e: relative relief; f: distance to fault; g: land cover; h: drainage density;

i: Peak Ground Accelation of the 2015 Gorkha earthquake (Peak Ground Acceleration data for the Gorkha earthquake

were obtained from USGS Shakemap, which was designed as a rapid response tool to portray the extent and variation of ground shaking throughout the affected region immediately following significant earthquakes); j: Average total monsoon precipitation (ICIMOD and the National Meteorological information Center of China. This data is the average precipitation for the period 1991-2010, for the monsoon season from June to October).

**Table 2** Susceptibility models for different triggers and landslide size classes in the Koshi River basin

| Landslide type | $x_1$ | $x_2$ | $x_3$ | $x_4$ | $x_5$ | $x_6$ | $x_7$ | $x_8$ | $x_9$ | $x_{10}$ | p |
|---|---|---|---|---|---|---|---|---|---|---|---|
| All RTL | - 6.4317 | 6.4955 | -12.2440 | - 0.1717 | -3.7048 | -1.3431 | 1.0590 | -0.7090 | 1.3725 | 0.7206 | 4.3961 |
| Small size RTL | - 8.36420 | 6.33158 | -1.37934 | - 0.09899 | -2.68158 | -1.91514 | 1.10489 | -0.93464 | 1.10003 | 0.98897 | -0.54775 |
| Large size RTL | - 4.93126 | 6.47043 | 7.03034 | - 0.30706 | 4.79661 | -0.13525 | 1.49649 | -0.49201 | 1.31034 | 0.07492 | -6.69787 |
| All ETL | -3.3342 | 5.8510 | -8.6844 | -0.5513 | 8.8514 | 6.3296 | 3.2108 | -0.2472 | 1.3740 | 17.4360 | -6.4566 |
| Small size ETL | -7.4433 | 5.8410 | -7.5233 | -0.1974 | 5.9871 | 4.2647 | 2.6977 | 1.7495 | 1.2858 | 7.5676 | -3.3845 |
| Large size ETL | 6.939 | 10.116 | -26.355 | 3.660 | 16.503 | 11.678 | 3.962 | -4.039 | 2.633 | 28.199 | -11.445 |

ROC curves were drawn to verify the accuracy of each susceptiblity model (Fig. 8), and the Area Under Curve (AUC) was calculated. The AUC values of the ETL models were higher than for RTL, since the ETL were more concentrated than the RTL, as the inventory is from one single triggering event, whereas the RTLs are from many different rainfall events over a longer time period.

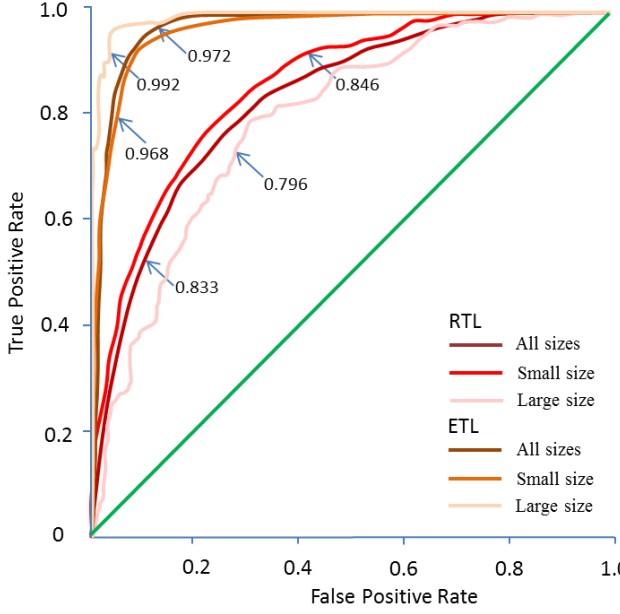

**Fig. 8** ROC curves for the susceptibility assessing models to different sizes of RTL and ETL

315

## 6.2 Results

The logistic regression models were employed to the Koshi River basin and in total six susceptibility maps were generated (Fig. 9). Susceptibility values were classified into four levels: low, moderate, high and very high, based on the following susceptibility threshold values: 0-0.25, 0.25-0.5, 0.5-0.75 and 0.75-1.

The RTL susceptibility map (Fig. 9a) shows that high and very high susceptible are located mostly in the Siwaliks and in the Mahabharat Lekh region in west-eastern direction and the Middle to High Himalaya region in north-south direction. The Siwaliks and Mahabharat Lekh regions (Fig. 1) have high and very high susceptibility levels for small landslides, and lower susceptibility levels for large ones. The Middle and High Himalaya region (Fig. 1) has a reverse situation: high and very high susceptibility levels for large landslides, and lower levels for small ones.

The ETL susceptibility map reflects the co-seismic landslide pattern of the Gorkha earthquake, with very high and high susceptibility in the western part of the Koshi River basin. It is important to note that the ETL susceptibility map only reflects the characteristics of the Gorkha earthquake and is therefore not a reliable map for future earthquakes that may have another epicentral location, length of fault ruptures and magnitudes.

Both ETL and RTL susceptibility maps show different patterns for the large size landslide class (Fig. 9 c and f), whereas the maps for small size (Fig. 9 b and e) resemble those of all size classes (Fig. 9 a and d). This is due to the relative small fraction of the large size landslides in comparison with the small ones, and their more restricted location, which gives different weight values for some factor maps (Table 2).

The highest susceptibility zones for small size and large size RTL show a large overlapping area, although the area of these classes is much smaller for large size RTL. In the Siwaliks and Mahabharat Lekh regions high and very high susceptibility zones for large size RTL are located in the upper steep hillslopes. In the Middle and High Himalaya region, the highest susceptibility zones for both small size and large size RTL are mostly located on steep slopes along rivers. The highest susceptibility zones for both small and large size ETL are located in the northwestern part of the Koshi basin. For large size ETL these are concentrated in a smaller area to the northeast of Kathmandu (with altitude higher than 3000m) where small ETL also show high susceptibility in the southeast of Kathmandu.

340

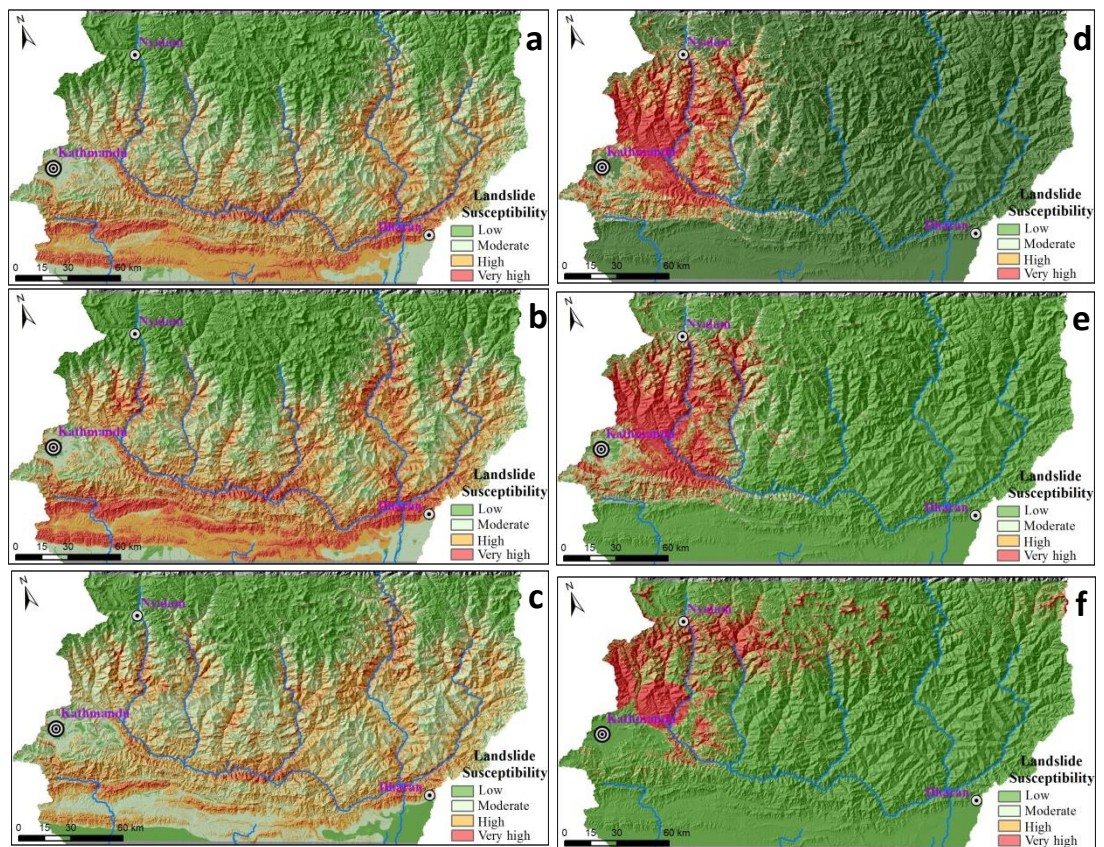

**Fig. 9** Susceptibility maps for different sizes of RTL and ETL: (a) for all RTLs; (b) for small RTLs; (c) for large RTLs; (d) for all ETLs; (e) for small ETLs; (f) for large ETLs.

The areal coverage of the landslide susceptibility classes was calculated for each susceptibility map (Fig. 10). Compared to RTL, the ETL susceptibility maps have a larger area with low susceptibility, due to fact that the Koshi River basin is far from the epicenter of Gorkha earthquake, thus the earthquake affected region is only part of the basin. The very high and high susceptible region for ETL is mostly concentrated in the western and southwestern parts of the basin, clearly reflecting the PGA pattern (Fig. 7i). The RTL susceptibility also reflects the triggering factor (monsoonal rainfall), with the highest susceptibility in the south of the basin. However, the higher rainfall peak in the Middle and High Himalaya region is less pronounced in the susceptibility maps, as well as in the inventory maps (Fig. 4). The higher susceptibility classes for large ETL occupy more area than for small ETL, while the opposite can be observed for RTL.

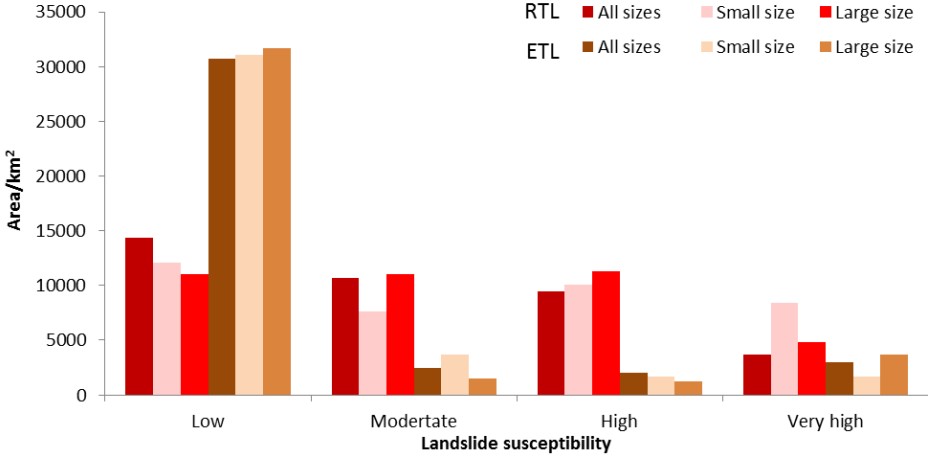


**Fig. 10** Coverage of different landslide susceptibility classes for ETL and RTL maps

**7. Validation of landslide susceptibility maps**
Different groups of landslide data were used to validate the landslide susceptibility maps for RTL and ETL. For each
trigger and size class, the number of landslides was calculated, inside the areas with a certain susceptibility level, to
cross-validate the results.

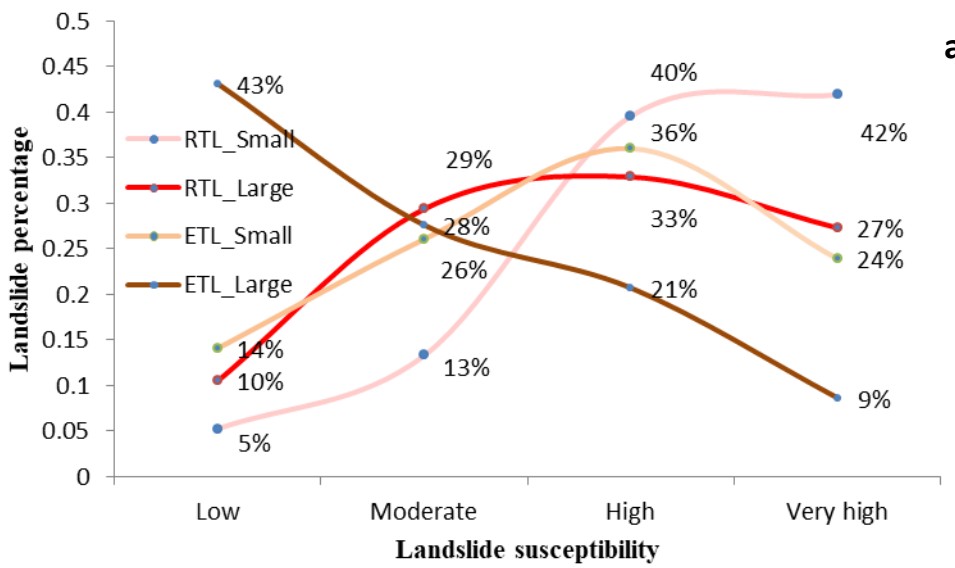


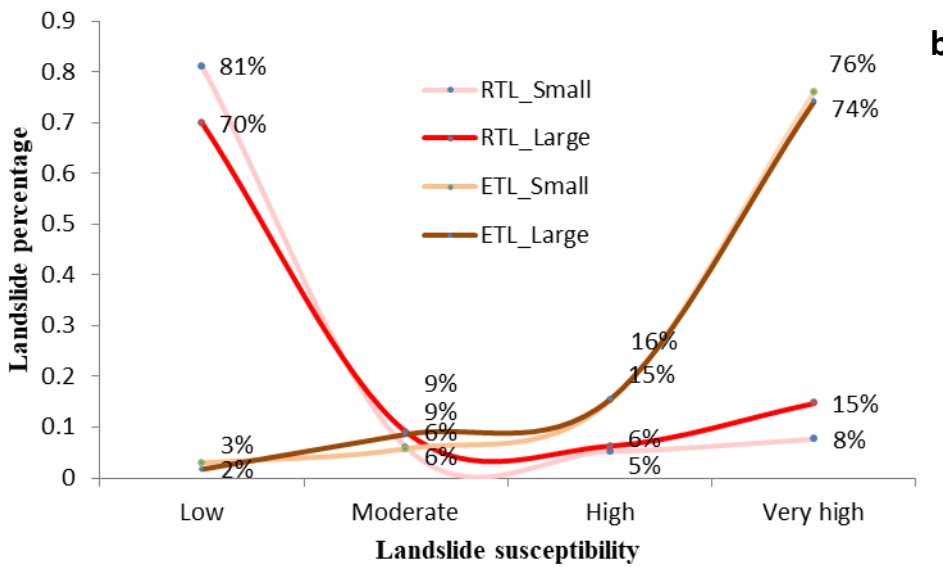

**Fig. 11** Cross validation of the landslide susceptibility maps. (a) The percentage of landslides in the various classes of the RTL susceptibility map; (b) The percentage of landslides in the various classes of the ETL susceptibility map.

The percentages of different size RTLs and ETLs in each susceptibility are shown in Fig.11. For the RTL susceptibility map, percentages of of small size and large size landslides show a similar tendency, for both triggers. Most of the landslides were located in high and very high susceptibility zones. Only large size of ETL shows an opposite tredency. There is a marked difference between the percentages of ETL and RTL in the ETL landslide susceptibility classes. the RTL and ETL percentages show completely different patterns. Most of the RTLs (both small and large) are located in the low ETL susceptible regions. Conversely, a large fraction of small size and large size of ETLs are located in the high susceptible regions.

## 8. Discussion

This study aimed to analyze independent rainfall- (RTL) and earthquake-triggered landslide (ETL) inventories for a large mountainous watershed in the Himalayas, located in India, Nepal and China. It is important to mention, that the two rainfall-triggered landslide inventories are not event-based inventories (Guzzetti et al., 2012 ). A major limitation in this work was that we were not able to use separate event-based inventories for RTLs, and only one event-based inventory for ETL. The collection of event-based inventories, both for rainfall and earthquake triggers, remains one of the main challenges in order to advance the study of landslide hazard at a watershed scale. Another limitation for this landslide inventory was that, the temporal and quality of high resolution images, as well as revegetation affects the number of historic landslide inventory. More and more researchers make great efforts on the event-based landslide

inventories and database (Marc et al., 2018), which may supply more samples for comparison studies of RTL and ETL. The two RTL inventories differ in the sense that the 1992 inventory is based on landslides that were large enough to be mapped on the topographic map, where as the inventory between 1992 and 2015 represents the landslides that could be mapped from multi-temporal images over a number of years. Both inventories were lacking a separation into initiation and accumulation parts, and no separation in landslide types could be made. The effects of amalgamation of landslides might certainly have played a role in the Frequency Area Distribution (Marc and Hovius, 2015) although we are not able to quantify this, due to lack of an independent dataset. For the 1992-2015 dataset we were able to control this as we carried out the image interpretation ourselves, but the pre-1992 inventory could not be verified as the aerial photographs that were used to generate the updated topographic maps, were not available to us. Although the two inventories differ substantially with respect to the number of small landslides, it is striking to see that the cut-off values, and β values in the Frequency Area Distribution (FAD) are similar. It is very difficult to obtain a complete event-based landslide inventory for rainfall triggered landslides in Nepal, as landslides are generally generated by a number of extreme rainfall events during the monsoon, which can not be separated, as the area is cloud-covered through most of the period. The earthquake triggered landslide distribution is an event-based inventory, for a single earthquake (2015 Gorkha) and based on an extensive mapping effort by Roback et al. (2017) resulting in an inventory that can be considered as complete (Tanyas et al., 2017a). When comparing the FAD for RTL and ETL it is striking that the size-frequency distributions for both ETL and RTL show very similar behaviour for landslides above the cut-off value of 30,000 $m^2$. Although there is no consensus regarding the factors dictating the power-law distribution of landslides, there is an accumlating evidence that topography, as well as mechanical properties, has to be one of an important controlling factors (e.g., Stark and Guzzeti, 2009; ten Brink et al., 2009; Frattini and Crosta, 2013; Liucci et al., 2017). Our finding regarding similar cutoff values obtained from different inventories created for the same area is also supporting this argument. This conclusion also supported by Marc et al., 2019, who found that similar Beta values between ETL and RTL, but the cutoff value is much smaller because a correction to remove runout was applied.

## 9.   Conclusions

The pattern of the triggers (precipitation in the Monsoon for RTL, and PGA distribution for ETL) have major influence on the distribution of landslides and susceptibility zones. These trigger patterns differ substantially. When moist airflow from the India Ocean crosses over the Mahabharat Lekh, the intensity of precipitation reduces because the altitude lowers and temperature rises. As the airflow continues northwards to the Middle Mountains and Transition Belt, it rises again and consequently induces high precipitation in the area at an altitude between 2500~4000m. It results in two high precipitation regions during the monsoon season (Fig.7 i), which are reflected in the zones of high susceptibility to RTL. The precipitation pattern is different from the PGA distribution (Fig.7 j) for the Gorkha

earthquake, with strong shaking area located in the North and North east of Kathmandu, with PGA values larger than 0.44g. One limitation need to be clarified that, normally the rainfall on the day of the land sliding event and antecedent daily rainfall, which have close correlation with landslide occurrence, are usually taken as the key factors for landslide threshold. But in this study the mean precipitation during monsoon season were taken as the rainfall factor. It could be only supply a general tendency for landslide distribution in regional scale. In the RTL susceptibility assessment model, the weight of precipitation factor is low, which means this factor was not strong correlated with landslide susceptibility. It is better to characterize the variability of daily rainfall during the monsoon season, and take into account the daily rainfall instead of the mean. So use the short-term rainfall variability to study the long term historical landslide inventory and susceptibility assessment may be more reasonable (Deal et al., 2017).

The distribution of RTL and ETL susceptibility classes are also very different. As the ETL susceptibility map is based on a single event, the distribution of the susceptibility classes is controlled by the PGA for the 2015 Gorkha earthquake, and the patterns of the ETL susceptibility map differs from the RTL susceptibility map. This was confirmed by the cross validation (Fig. 11), which showed that the RTL susceptibility map has a modest capability of explaining the ETL pattern, but that the ETL susceptibility cannot properly predict the RTLs.

This means one should be careful with using susceptibility maps that were made for earthquake induced landslides, as prediction tools for rainfall induced landslides. Such maps are in fact of little practical implication, as the next earthquake may not be likely to occur in the same location and therefore produce a similar landslide pattern. The generation of ETL susceptibility maps should not be based on single earthquake scenario scenarios (Jibson, 2011), and ideally many earthquake scenarios should be used to model the overall ETL susceptibility. However, using PGA values based on probabilistic seismic hazard assessment might result is relatively poor statistical correlations with event-based inventories. Therefore, PGA maps and ETL inventories of specific earthquake scenarios are required to improve the statistical models. This requires more event-based ETL inventores, and efforts to generate worlwide digital databases should be encouraged (Tanyas et al., 2017a).

The relationship between ETL and RTL might also change over time. Rainfall-induced landslide activity is generally much higher in the first years after an earthquake, and generally decreases to pre-earthquake levels within a decade, due to depletion of co-seismic sediments, progressive coarsening of available sediments and revegetation (Fan et al., 2018b; Hovius et al., 2011; Marc et al., 2015). Landslide susceptibility map should also be updated after major earthquakes.

Both ETL susceptibility maps and RTL susceptibility maps show different patterns for large landslides, as compared to the small landslide or all landslides. In general the susceptibility maps, for both RTL and ETL, for all landslide sizes together show a large similarity with the ones for the small landslides only. This is due to the fact that the number of large landslides is quite limited as compared to the small landslides (See Table 1), and the samples used for generation

the models for all landslides and only small landslides are almost the same. However, the resulting susceptibility patterns are quite different, and it is therefore questionable whether landslide susceptibility maps that are generated for all landslide size would be able to accurately predict the large landslides. More emphasis should be given to the evaluation of landslide size in susceptibility and subsequent hazard and risk assessment. This is relevant for analyzing the potential runout areas of landslides and for evaluation landslide damming susceptibility (Fan et al., 2014; 2018b). Therefore, size and trigger matter in landslide susceptibility assessment.

## 10. Acknowledgements

This research was supported by the National Natural Science Foundation of China (Grant No.41401007), the External Cooperation Program of BIC, Chinese Academy of Sciences (Grant No. 131551KYSB20130003) and the "135" Program of IMHE (Grant No. SDS-135-1708). This study was also jointly supported by the Australian government funded Koshi Basin Programme at ICIMOD as well as ICIMOD's core funds contributed by the governments of Afghanistan, Australia, Austria, Bangladesh, Bhutan, China, India, Myanmar, Nepal, Norway, Pakistan, Switzerland, and the United Kingdom.

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
