# Peer review of "How size and trigger matter: analyzing rainfall- and earthquake-triggered landslide inventories and their causal relation in the Koshi River basin, Central Himalaya"

_Natural Hazards and Earth System Sciences, 2018_

## Short Comment (SC1) · 20 Jun 2018

Zhang et al. present a study in which they analyse rainfall- and earthquake-triggered landslide inventories to seek similarities, differences and correlations in regards of their frequency-area distributions, size ranges, controlling factors, and ultimately their susceptibility assessments. I find this study very interesting as it constitutes a systematic and practical-oriented regional-scale analysis of landslide patterns with recognized different triggers. I believe this study can promote further investigations by the scientific community in earthquake-prone mountainous areas for which detailed inventories are

available, which may confirm some of the authors' general observations and at the same time highlight some region-specific patterns.

I have just some observations and suggestions regarding some parts of the methodology, which are detailed below:

line 145 - I understand that the inventories were made through visual interpretation. It would be good if the authors specify this here rather than at line 150 (which refers only to the most recent images). Furthermore, it would be good to specify if and how the authors evaluated the mapping uncertainties due to low imagery resolution and visual interpretation, for instance in terms of shape and size mismatch and amalgamation, and their propagation to landslide statistics (e.g.frequency-area distributions, classification by controlling factors).

line 168 - Also here, it would be good to specify how the rather low spatial resolution of the DEM (30x30 m) affects the classification especially of landslides with small area (as low as 50 sq.m).

line 176 - Here it would be nice to explain the 60%-40% choice (is it because of the sample size? is it arbitrary?) and to specify how the landslides are assigned to either set (e.g. randomly, but being sure that the size distribution and controlling factors classification are the same in both sets?).

line 216 - Here you classify the landslides into small and large depending on "field experience" and on the basis of the frequency-area distributions. You choose 6000 m2 as your threshold which is more or less the cut-off value in the frequency-area distribution of the earthquake-triggered landslides but is much smaller than that of the rainfall-triggered landslides. However, the cut-off (or rollover point) may be affected by undersampling of small landslides, which you should be able to rule out explicitly. Also, what field experience means in this context remains unclear. So, this threshold area seems quite arbitrary. I would encourage the authors to introduce a physically-based justification for this choice, which you did in part already in the introduction. On the

other hand, I would also suggest that you run your model multiple times with different thresholds, to show if there is an optimal (data-driven) threshold that can best differentiate the statistics of RTL and ETL in your study area. This threshold will certainly have a hidden physical meaning, which could be then discussed.

Gianvito Scaringi

---

## Referee Comment (RC1) · O. Marc (Referee) · 9 Jul 2018

The work by Zhang et al., presents two landslide inventories obtained with satellite and aerial imagery over a large catchment (Koshi River) in the central Himalaya. One inventory contains rainfall triggered landslides (RTL) as observed in 1992 and in 2015, while the other contains earthquake-triggered landslide (ETL) from the 2015 Gorkha earthquake. They compare both dataset relative to landscape properties (altitude, slope gradient and aspect, soil cover, etc) and derive a susceptibility model from each inventory to assess whether both susceptibility model agree and can be inter-changed or

not. The authors also propose a size cutoff and performs various analysis for landslide smaller or larger than this cutoff. Overall I think some of the question discussed in the paper may be worth to be investigated and published within NHESS, however the current manuscript is unclear or lacking details in many places, and for me the 2 main results stated by the authors ("size and trigger matter") are poorly supported by their current analysis. In depth revisions are clearly needed in my opinion, and I propose below several directions to clarify and improve the analyses. Whether or not the authors claims will stand after these re-analysis is unclear as of now.

Please find my detailed review within the attached document.

Please also note the supplement to this comment:
https://www.nat-hazards-earth-syst-sci-discuss.net/nhess-2018-109/nhess-2018-109-RC1-supplement.pdf
* * *
[Figure]

**Supplement:**

Review of Zhang

The work by Zhang et al., presents two landslide inventories obtained with satellite and aerial imagery over a large catchment (Koshi River) in the central Himalaya. One inventory contains rainfall triggered landslides (RTL) as observed in 1992 and in 2015, while the other contains earthquake-triggered landslide (ETL) from the 2015 Gorkha earthquake. They compare both dataset relative to landscape properties (altitude, slope gradient and aspect, soil cover, etc) and derive a susceptibility model from each inventory to assess whether both susceptibility model agree and can be inter-changed or not. The authors also propose a size cutoff and performs various analysis for landslide smaller or larger than this cutoff.

Overall I think some of the question discussed in the paper are worth to be investigated and published within NHESS, however the current manuscript is unclear or lacking details in many places, and for me the 2 main results stated by the authors (size and trigger matter) are poorly supported by their current analysis. In depth revisions are clearly needed in my opinion, and I propose below several directions to clarify and improve the analyses, whether or not the authors claims will stand after these re-analysis is unclear as of now.

**Major comments**

1/ The authors present new inventories but there is a lack of description of mapping : what about amalgamation of landslides (cf Marc and Hovius 2015)? What about the mapping of debris flow ? etc What about the mapping resolution effects on the size distribution roll-over? With airphotos and Google Earth what was the highest altitude where comprehensive mapping could be performed ?
Also I think a brief comparison of the ETL mapped by the authors with the public dataset of Roback et al., (2017) would be useful to validate mapping.

2/ In the introduction the authors state that susceptibility comes from Internal and External factor, but later you use no external factor for Rainfall.  This is a problem I would say because your susceptibility maps for EQIL and RIL have all internal parameters in common, so it is a bit as if you assumed rainfall forcing was homogeneous across the study area, while it is not. I thnik it would be worth to try to constrain your RIL with a long term average pattern of Rainfall (i.e. climatologic mean summer rainfall ?). This can exactly be done with a TRMM climatology, as presented by Bookhagen and Burbank 2006. Other option may also be possible. This would be a great improvement for the paper, and should be at least mention and discussed.
        In any case, the comparison of the two susceptibility model does not necessarily depends on the different trigger but very possibly on the relevant landscape properties, as the coverage zone for the two model are very different. I strongly think that this possibility needs to be quantitatively assessed before possible publication.

3/ The author spend quite some time discussing size-effects in the introduction and in their analysis, but their is almost no explanation on how they choose/find their threshold  for small or large landslide size. Second : In Fig  5, 6 and 7 (and maybe 8 at least for ETL) there is nothing that strongly suggest any significative difference between small and large landslize. The statement that "size matters" in the title, abstract and conclusions is for me completely unsupported. Further, I do not see really any place where the authors summarize in what  size would matter (in the result section) and why it could (at least in discussion ).

4/ I think the purpose of the paper and its relation to the state of the art literature is not very clearly

presented, and would suggest that the authors try to clarify several parts of the introduction (cf. Minor comments).

5/ The discussion and conclusions section is using vague or inaccurate formulations and is missing a lot of references ( there is only 1 on the rainfall pattern !!) on the importance of the seismic shaking pattern for example, on the elevated landslide susceptibility caused by loose landslide deposits or by slopes damaged by the shaking but unfailed. Potential model bias or difference in the mechanics of small or large landslides are also not discussed. Significant improvement are possible and needed (cf. Minor comments).

**Detailed comments**

L47 "To investigate whether earthquake- and rainfall-triggered landslides inventories have similar area-frequency distributions, area-volume relations and spatially controlling factors, it is important to collect event-based landslide inventories. The difficulty is to collect complete inventories that are independent for earthquakes and rainfalls. Many studies that compare the characteristics of earthquake- and rainfall-triggered landslide inventories focus on mapping landslides triggered by rainfall after major earthquakes."
>> The question underlying this study is unclear. The literature overview seems biased and inexact. Since decades they are indepedent rainfall inventories : New Zealand, Taiwan, Guatemala (Hovius 1997, 2000, Malamud, 2004) and others...
The study cited on L51-60 presumably looked at rainfall associated to EQ on purpose, to study whether or not an earthquake affected the properties of subsequent rainfall induced behavior.

L68 "There are fewer studies that compare the two triggering mechanisms in an independent manner."
Fewer ? Then cite them or say No studies. Malamud 2004 did. Meunier too. Again it is unclear in the introduction what the authr want to compare ? I recognize that there is a value into comparing rainfall and EQ induced landslide in the same area, to normalize for landscape properties. But if this is the aim of the authors this is not clearly stated.
I also do not see the problem of the study of Lin 2006 and Chang 2007 in Taiwan : They mapped rainfall landslide before the EQ exactly has the author are doing here.

L71-72: I am not sure "potential causal factor" are appropriate terms, given the trigger could also be considered a necessary term to "cause" the landslide. In-situ properties maybe although this is almost identical to internal factors...
I also note that from a physical point of view I would say that landslide occurrence is the convolution of a susceptibility term (due to in-situ/internal factor) and a forcing or triggering term. This may be the most adequate view point for a landslide event analysis (e.g. Meunier 2013, Barlow 2016).
From a probabilistic point of view, used for hazard analysis, the landslide susceptibility does not design the intensity of the response of a slope to a given forcing, but the long-term probability of landslide occurrence, including both in-situ properties, and the probability of various trigger. This is most suited for historical landslide inventories, where individual triggers are not or poorly constrained.
The authors do not really stick to one frame that makes the term susceptibility ambiguous in their study. Indeed in they state in essence in L70-71: Susceptibility (probabilistic sense) depends on internal factor (that makes area susceptible (physical sense)) and triggering factors.
This sentence and probably couple of others could be rephrased to avoid this ambivalent and possibly confusing uses.

L83 : "There is no clear evidence shows the difference on morphology between rainfall-triggered landslide and earthquake- triggered landslide"
>> Unclear statement. Could the authors specify what they mean with morphology ?
Also incorrect grammar :"that shows" or "showing"

L84 : also unclear. Rephrasing needed. Which statistics?

L92: huge slides ? Give a size range maybe.

L95: "whether it is possible to utilize inventories of earthquake triggered landslides (ETL) as inputs for analyzing the susceptibility of rainfall-triggered landslides (RTL)."
Depending on what authors means by the "susceptibility" here (cf comment above), the problem can be ill-posed given that obviously Rtl and ETL depends on a different trigger and thus will likely show different patterns (as shown by other studies: Meunier et al., 2008, Marc et al., 2018)

L151: It is unclear what you did with Landsat and ASTER DEM. ? Map or only adjust locations of landslides mapped with Google Earth or topo maps? The use of "therefore" is confusing.
The author should precise (in Fig 1?) where Topo maps where used and where Google Earth. With overlap or not ? Is the mapping style in topo maps consistent with Google Earth ?

L155 : resolution of satellite ?

L159 : Confusing sentence, clarify or rewrite.

L160 : consider replacing "rainfall impact to landslide" by something clearer, like : new or reactivated landslide due to subsequent rainfall.

L161 : which pre EQ image ? Google Erth or other... Estimation of the areas where pre or post EQ imagery did not allow mapping ( because of clouds or shadows)

L164 : You said above you did not separate different zones of the andslides. How did you choose where was the initiation point? Is it the hghest point? Taking a single pixel as source or scar zone may bias your statistics. Why not considering a scar surface in the upper part of the polygon ?

L166 : Line 151 you said you use ASTER GDEM ( 30m). Be consistent. There is absolutely no reason to use a 90m dem while SRTM 30m is available. For quantitative slope assessment it will make a difference and analysis should be re performed with the highest possible resolution.

L167 : Explain how you determine where the river network start, as this is not done by arc GIS.

L172: you mean it is from Shakemap ? At which resolution ? In any case a few sentences on how shakemaps are derived and on what are their limitations ( no topographic amplification, no constraints on site effects within mountainous area, interpolation with heavy weight given to station measurements even in areas with very different setting ) is needed, together with a couple of references. I also think a map of the shaking in the Koshi, with landslides indicated, should be shown at least in supplement.

L183: Did you use distance to river ( as suggested above) or not ? What is relative relief , computed at which scale ? Same drainage density ? Distance to fault, which faults ? I think a supplementary figure

with the different (relevant) susceptibility factor would be useful.

L207-210: which method did you use to determine the Beta exponent and the threshold size ? Clauset et al. 2009 is the recommended approach (and they provide script to reproduce their analysis). Are the different estimates significatively different (i.e., what are the uncertainty on them)? ETL-All and the two RTL dataset have very close exponents.

P9 214: landslide size definition : is there a mistake or this classification is discontinuous ? small <1000 ; 1000< medium <10,000 ; <100,000 large...
What about landslide between 10,000 and 100,000??

L216: why 6000? you say it is based on FAD but without explanation... The sentence above is meaning less, which FAD analysis ? Which field exp ?6000 i the power-law cutofffor ETL but is in the roll over of RTL....
Also a few sentence on the meaning of the roll-over (and its sensitivity toresolution censoring) and of the Beta exponent and how it may be linked to physical properties is needed ! Cf Pelletier 1997, Stark and Hovius 2001, Stark and Guzzetti 2009, Frattini and Crosta 2013,

L224: For this initial correlation did you use ETL or only RTL ? If ETL was used what about PGA ?

L229-231: I am not sure this comparative analysis in terms of altitude or other parameters make any sense : because the difference will not have any thing to do with EQ or Rain , just to the fact that one dataset (RIL) covers 10-20 times more area, with a vast area at low elevation. Instead the ETL are limited, because of the fault location, to a small zone with high elevation.
I think all this analysis should be redone : ETL and RTL should be comapred to the landscape within which they occur, so that it i not absolute elevation or slope or aspect that is analyzd but fraction of the lanscape (percentile of landscape elevation for exemple, or analysis of oversampling or undersampling of given slopes or aspect. Cf Meunier 2008, Barlow 2016 etc). Fig 5 should also be updated.

L234: Is this based on the land cover maps ? Or is this from the imagery ?

L244: Missing word...to the?? direction ?

L264: gully density ? Or drainage density ? Be consistent !

L267-269: Could you comment on the values given for the different model ? It reaches 24 / 22 for ETL against 7 /6 for RTL. The methods sequence could include some more details to allow the author to have an intuition about the relative importance of different parameters.

L284: Obviously landslide susceptibility of ETL is giving only high suscptibility where you had data...
As mentionned above you should also show the Shaking map ...

L289 : EQ without effect on large landslides ? The argument that large landslides occur only close the fault may be true for very large landslides but seems unikely for landslides down to 6000m2 that is not so large.

L335 : I thnk this conclusion is erroneous, or at least not demonstrated by the authors. Because the ETL model includes PGA, and also because it is based on a much smaller part of the landscape, a subarea where landslide are located in a different environent copared to the zone affected by RTL.

I think only by limiting the model development in an area where both RTL and ETL are widespread could the authors try to test this hypothesis.

L340: You repeat this result that is completely obscure in the main text. There was no reason given to this threshold value.

L342 : You never demonstrated the correlation in altitude and aspect was due to precipitation... The following sentences are interesting but a bit weak. The use of some rainfall climatology ( as existing with TRMM for example, would be an actual demonstration).

L349: Should be rephrased. The epicenter is extremely far from your study area and seismic waves propagates in all directions. Second part may refer to seismic directivity that relates to wave interference. I think a discussion in terms of the ground motion pattern is what you mean. (and it is difficult to discuss without showing the shaking in a figure...)

L365: The forcing extent are different within this catchment. You need to discuss it, and for that you need to show shaking and rainfall pattern, both essential information that are missing !

L368: "Some more detail information could be included in large scale research"
>> Like what ? why would it help? and why didn't you include it ? As of now this sentence does not bring anything to the reader.

L376: "Whereas, the use of rainfall-triggered landslide maps can be of some use for predicting the occurrence of earthquake-triggered landslides, one should be careful, as the specific location of the earthquake plays a dominant role."
>> Not sure Whereas is the proper word. Anyway I do not think there is anything new for the community in a sentence like that.

Fig 5 : Why show Altitude vs other param? This display does show nicely the difference in altitude between datasets but not really with the other parameters.
Further it is hard to interpret anything when the distribution of landscape parameters is not shown... I think the authors must show the distribution of landscape properties (as classically done in the literature) slope gradient, aspect, altitude; stratum etc in the study area in black and then the ones of landslide RTL/ ETL in red / yellow on top for comparison.

Fig 6 : I suggest that you put all RTL in the left column and ETL in the right. It will make the figure less confusing and subplots easier to compare.

Fig 8 : large RTL are better predicted. Do you think this is physical or it may be a bias due to the higher difficulty to map small landslides? Also  is there any ROC difference between RTL at any size and the 1992 or 2015 inventories ?

Fig 9  : Comparison is not ideal : ETL susceptibility is likely driven by the fact landslides are limited to a very small subset of the Koshi.

**Technical comments:**
L85/86 : "are" missing between volume smaller/larger

L97 -> hazard and risk assessment (i.e. remove 1 assessment)

L214: From the Biblio it should be Tong et al. 2013. Given that this is a book in Chinese I doubt that this references will be accessible by much reader... and not sure it is essential .

L 385 : Weather > Whether

Fig 7 caption : "Statistics" ;  susceptiblity x2 > missing "i"

Overall the phrasing is a bit awkward or erroneous in many more places than reported here, and I invite the authors to carefully read again the manuscript.

**References used in the review and not in the study:**

Barlow, J., Barisin, I., Rosser, N., Petley, D., Densmore, A. and Wright, T.: Seismically-induced mass movements and volumetric fluxes resulting from the 2010 Mw = 7.2 earthquake in the Sierra Cucapah, Mexico, Geomorphology, doi:10.1016/j.geomorph.2014.11.012, 2014.

Frattini, P. and Crosta, G. B.: The role of material properties and landscape morphology on landslide size distributions, Earth and Planetary Science Letters, 361, 310–319, doi:10.1016/j.epsl.2012.10.029, 2013.

Hovius, N., Stark, C. P. and Allen, P. A.: Sediment flux from a mountain belt derived by landslide mapping, Geology, 25(3), 231–234, doi:10.1130/0091-7613(1997)025<0231:SFFAMB>2.3.CO;2, 1997.

Hovius, N., Stark, C. P., Hao-Tsu, Chu and Jiun-Chuan, L.: Supply and Removal of Sediment in a Landslide-Dominated Mountain Belt: Central Range, Taiwan, The Journal of Geology, 108(1), 73–89, doi:10.1086/jg.2000.108.issue-1, 2000.

Malamud, B. D., Turcotte, D. L., Guzzetti, F. and Reichenbach, P.: Landslide inventories and their statistical properties, Earth Surf. Process. Landforms, 29(6), 687–711, doi:10.1002/esp.1064, 2004.

Marc, O., Stumpf, A., Malet, J.-P., Gosset, M., Uchida, T. and Chiang, S.-H.: Towards a global database of rainfall-induced landslide inventories: first insights from past and new events, Earth Surface Dynamics Discussions, 2018, 1–28, doi:10.5194/esurf-2018-20, 2018.

Meunier, P., Uchida, T. and Hovius, N.: Landslide patterns reveal the sources of large earthquakes, Earth and Planetary Science Letters, 363, 27–33, doi:10.1016/j.epsl.2012.12.018, 2013.

Pelletier, J. D., Malamud, B. D., Blodgett, T. and Turcotte, D. L.: Scale-invariance of soil moisture

variability and its implications for the frequency-size distribution of landslides, Engineering Geology, 48(3–4), 255–268, doi:10.1016/S0013-7952(97)00041-0, 1997.

Stark, C. P. and Hovius, N.: The characterization of landslide size distributions, Geophysical Research Letters, 28(6), 1091–1094, doi:10.1029/2000GL008527, 2001.

Stark, C. P. and Guzzetti, F.: Landslide rupture and the probability distribution of mobilized debris volumes, J. Geophys. Res., 114(F2), F00A02, doi:10.1029/2008JF001008, 2009.

---

## Referee Comment (RC2) · Anonymous Referee #2 · 13 Jul 2018

[referee-annotated manuscript omitted]

---

## Author Comment (AC2) · 26 Sep 2018

Dear Dr. Odin Marc,

Thank you so much for your careful reviewing to the paper. Your comments and suggestions are very helpful to improve the quality of this paper. We considered about your questions one by one, and tried to modify the manuscript.

Some major work we have done to the paper: (1) Comparing the landslide inventories of Gorkha earthquake, and got one more compete inventory; (2) Base on the inven-

tories, the new threshold value for large and small size landslide was analyzed; (3) Frequency Ratio was employed to analyze the correlation between landslide and contributing factors; (4) DEM data was changed to higher resolution data, and precipitation factor was employed to the susceptibility model for RTL. (5) Introduction and discussion parts were rewritten.

We made a table to response the comments one by one, and also marked the changed part in the manuscript. Please see them in the supplement. Thanks again!

Best regards, Jianqiang Zhang

Please also note the supplement to this comment:
https://www.nat-hazards-earth-syst-sci-discuss.net/nhess-2018-109/nhess-2018-109-AC2-supplement.zip

---

## Author Comment (AC3) · 26 Sep 2018

Dear Reviewer,

Thanks a lot for your kindly reviewing to the paper. Many details were given in your comments. We tried to modified the paper according to your suggestions and comments.

(1) Higher resolution DEM, ALOS PALSAR DEM was brought in to the modeling;

[Figure]

(2) Some descriptions and references to R and ROC curve were added to the paper;

(3) We discuss more in the section of FAD part, to explain the threshold for large size and small size landslide.

(4) Many editing errors were modified in the paper.

Please see the details of changes in the supplement. We made table to response your comments one by one. And the revised copy of manuscript was attached.

Best regards, Jianqiang Zhang

Please also note the supplement to this comment:
https://www.nat-hazards-earth-syst-sci-discuss.net/nhess-2018-109/nhess-2018-109-AC3-supplement.zip

―――――――――――――――――

---

## Author Response (AR1)

**Response to RC 1**

| Comments | | Response |
|---|---|---|
| **Major comments** | | |
| 1 | The authors present new inventories but there is a lack of description of mapping: what about amalgamation of landslides (cf Marc and Hovius 2015)? What about the mapping of debris flow ? etc
What about the mapping resolution effects on the size distribution roll-over? With airphotos and Google Earth what was the highest altitude where comprehensive mapping could be performed? Also I think a brief comparison of the ETL mapped by the authors with the public dataset of Roback et al., (2017) would be useful to validate mapping. | Thank you for your answer. We have addressed the issue of amalgamation of landslides as one of the issue involved in analyzing area-size distributions in the introduction. We did not separate the landslides in erosional and accumulation areas, and therefore we are not able to analyze this effect quantitatively in this study.
Based on your suggestions we have decided to use the earthquake induced landslide inventory from Roback et al. (2017) as this was much more complete then the one we generated.

See Line 156-162 |
| 2 | In the introduction the authors state that susceptibility comes from Internal and External factor, but later you use no external factor for Rainfall. This is a problem I would say because your susceptibility
maps for EQIL and RIL have all internal parameters in common, so it is a bit as if you assumed rainfall forcing was homogeneous across the study area, while it is not. I think it would be worth to try to constrain your RIL with a long term average pattern of Rainfall (i.e. climatologic mean summer rainfall?). This can exactly be done with a TRMM climatology, as presented by Bookhagen and | Thanks for your suggestion. We agree that precipitation plays an important role in the occurrence of rainfall-triggered landslides. During our research we found that rainfall intensity has a stronger effect on landslide occurrence than long term precipitation, like annual precipitation. But due to the limitation of precipitation data in Nepal, we were not able to represent this spatially. Therefore we used a dataset representing the average precipitation during the monsoon season from ICIMOD and the National Meteorological information Center of China. This data is the average precipitation for the period 1991-2010, for the monsoon season from June to October. We used this dataset in the analysis, and adjusted the text, tables and figures accordingly . |

| | | |
|---|---|---|
| | Burbank 2006.
Other option may also be possible. This would be a great improvement for the paper, and should be at least mention and discussed. In any case, the comparison of the two susceptibility model does not necessarily depends on the different trigger but very possibly on the relevant landscape properties, as the coverage zone for the two model are very different. I strongly think that this possibility needs to be quantitatively assessed before possible publication. | See Line 287 Fig. 6 |
| 3 | The author spend quite some time discussing size-effects in the introduction and in their analysis, but their is almost no explanation on how they choose/find their threshold for small or large landslide size. Second : In Fig 5, 6 and 7 (and maybe 8 at least for ETL) there is nothing that strongly suggest any significative difference between small and large landslide. The statement that "size matters" in the title, abstract and conclusions is for me completely unsupported. Further, I do not see really any place where the authors summarize in what size would matter (in the result section) and why it could (at least in discussion ). | For defining the threshold of landslide size, we based ourselves on the area-frequency distribution analysis. We used the cut-off point, the point where the distribution starts to deviate from a power-law relation as the threshold value to differentiate between small and large landslides. The results showed that the cut-off points for the two rainfall induced and the earthquake induced inventories were quite similar, and a threshold of 30,000 $m^2$ was used. We modified the text and figures to incorporate this.

See line 211-229. |
| 4 | I think the purpose of the paper and its relation to the state of the art literature is not very clearly presented, and would suggest that the authors try to clarify several parts of the introduction (cf. Minor comments). | We have improved the introduction section and incorporated more literature to better represent the state of art and the issues related to the differences in earthquake and rainfall induced landslide inventories and susceptibility. We wanted to highlight that there actually very few studies that have compared susceptibility maps from different triggers in the same area, in an independent manner (So not |

| | | |
|---|---|---|
| | | specifically post-earthquake rainfall induced landslides), and also the limited role of landslide size in landslide susceptibility modeling.

See Line 58-73. |
| 5 | The discussion and conclusions section is using vague or inaccurate formulations and is missing a lot of references ( there is only 1 on the rainfall pattern !!) on the importance of the seismic shaking pattern for example, on the elevated landslide susceptibility caused by loose landslide deposits or by slopes damaged by the shaking but unfailed. Potential model bias or difference in the mechanics of small or large landslides are also not discussed. Significant improvement are possible and needed (cf. Minor comments). | We have now rewritten the discussion and conclusions section and we added a number of relevant references. |
| | | |
| | **Detailed comments** | |
| 1 | L47 "To investigate whether earthquake- and rainfall-triggered landslides inventories have similar area frequency distributions, area-volume relations and spatially controlling factors, it is important to collect event-based landslide inventories. The difficulty is to collect complete inventories that are independent for earthquakes and rainfalls. Many studies that compare the characteristics of earthquake- and rainfall triggered landslide inventories focus on mapping landslides triggered by rainfall after major earthquakes." | There are undoubtedly many independent inventories of earthquake and landslide triggered landslides available, but rather few that come from the same study area. Even more so independent inventories that are not rainfall induced landslides inventories in the years after an earthquake. Ideally one would like to have several complete landslide inventories produced by rainfall events with different return periods, and several earthquake induced landslide inventories produced by different earthquake scenarios in the same study area. So we do not want to study the post-earthquake |

| | | |
|---|---|---|
| | >> The question underlying this study is unclear. The literature overview seems biased and inexact. Since decades they are indepedent rainfall inventories : New Zealand, Taiwan, Guatemala (Hovius 1997, 2000, Malamud, 2004) and others... The study cited on L51-60 presumably looked at rainfall associated to EQ on purpose, to study whether or not an earthquake affected the properties of subsequent rainfall induced behavior. | |
| 2 | L68 "There are fewer studies that compare the two triggering mechanisms in an independent manner." Fewer? Then cite them or say No studies. Malamud 2004 did. Meunier too. Again, it is unclear in the introduction what the author want to compare? I recognize that there is a value into comparing rainfall and EQ induced landslide in the same area, to normalize for landscape properties. But if this is the aim of the authors this is not clearly stated. I also do not see the problem of the study of Lin 2006 and Chang 2007 in Taiwan : They mapped rainfall landslide before the EQ exactly has the author are doing here | Thank you for your comment. We have tried to make a more clear that the main aim of this study is to compare how earthquake and rainfall triggered landslide inventories lead to different susceptibility maps, and that also different landslide size classes have different causal factor combination and lead to different susceptibility maps.

See line 74-79. |
| 3 | L71-72: I am not sure "potential causal factor" are appropriate terms, given the trigger could also be considered a necessary term to "cause" the landslide. In-situ properties maybe although this is almost identical to internal factors...
I also note that from a physical point of view I would say that landslide occurrence is the convolution of a | We have adjusted this in the text. We agree with the observation that the susceptibility takes into account the spatial patterns of contributing factors and triggering factors. Landslide inventories for specific earthquake and rainfall events are required to estimate the landslide density for specific return periods. |

| | | susceptibility term (due to in-situ/internal factor) and a forcing or triggering term. This may be the most adequate view point for a landslide event analysis (e.g. Meunier 2013, Barlow 2016). From a probabilistic point of view, used for hazard analysis, the landslide susceptibility does not design the intensity of the response of a slope to a given forcing, but the long-term probability of landslide occurrence, including both in-situ properties, and the probability of various trigger. This is most suited for historical landslide inventories, where individual triggers are not or poorly constrained. The authors do not really stick to one frame that makes the term susceptibility ambiguous in their study. Indeed in they state in essence in L70-71: Susceptibility (probabilistic sense) depends on internal factor(that makes area susceptible (physical sense)) and triggering factors. This sentence and probably couple of others could be rephrased to avoid this ambivalent and possibly confusing uses. | |
|---|---|---|---|
| 4 | L83 : "There is no clear evidence shows the difference on morphology between rainfall-triggered landslide and earthquake- triggered landslide"
 >> Unclear statement. Could the authors specify what they mean with morphology ?
 Also incorrect grammar :"that shows" or "showing" | We have adjusted this in the text, and modified the introduction |
| 5 | L84 : also unclear. Rephrasing needed. Which statistics? | We have adjusted this in the text, and modified the introduction |

| 6 | L92: huge slides ? Give a size range maybe. | We have adjusted this in the text, and modified the introduction |
|---|---|---|
| 7 | L95: "whether it is possible to utilize inventories of earthquake triggered landslides (ETL) as inputs for analyzing the susceptibility of rainfall-triggered landslides (RTL)." Depending on what authors means by the "susceptibility" here (cf comment above), the problem can be ill-posed given that obviously Rtl and ETL depends on a different trigger and thus will likely show different patterns (as shown by other studies: Meunier et al., 2008, Marc et al., 2018) | Many landslide susceptibility maps are generated by making a statistical relation between landslide occurrences and contributing factors. There are many instances where there are no separate inventories available for individual triggering events, and where it is not possible to separate landslides triggered by earthquakes from landslides triggered by rainfall. If a susceptibility map that was generated from multi-temporal landslides is used as the basis for hazard and risk assessment and land use zoning, it might result in very wrong predictions in case of an earthquake. And vice versa, if an earthquake induced inventory is used as the basis for a landslide susceptibility for the period after, say a decade, it might also be quite wrong. Furthermore we also address in this research that apart from the trigger, also size matters. |
| 8 | L151: It is unclear what you did with Landsat and ASTER DEM. ? Map or only adjust locations of landslides mapped with Google Earth or topo maps? The use of "therefore" is confusing.
The author should precise (in Fig 1?) where Topo maps where used and where Google Earth. With overlap or not ? Is the mapping style in topo maps consistent with Google Earth ? | Our description was not clear. We changed the sentence to *Images from Google Earth were downloaded and geo-referenced and landslides were mapped using visual image interpretation and screen digitizing* |
| 9 | L155 : resolution of satellite ? | For this paragraph, we changed the method and description. After the 2015 April 25th Gorkha earthquake, earthquake-triggered landslides were mapped by Roback et al.(2017) using high-resolution (<1m pixel resolution) pre- and post-event satellite imagery. 24,915 landslide areas were mapped, and 1,4000 landslides were distributed in Koshi river basin. Chinese GaoFen-1 and GaoFen-2 satellites imageries (with 2.5m resolution) of the CNSA (China National Space |

| | | |
|---|---|---|
| | | Administration), which are part of the HDEOS (High-Definition Earth Observation Satellite) program, were employed to validate this landslide inventory. These images were captured during 27 April, 2015 to May 14 2015. Finally 15 landslide areas were deleted, and 120 landslide areas were added to the inventory. |
| 10 | L159 : Confusing sentence, clarify or rewrite | We have rewritten this sentence. |
| 11 | L160 : consider replacing "rainfall impact to landslide" by something clearer, like : new or reactivated landslide due to subsequent rainfall. | We have adjusted this |
| 12 | L161 : which pre EQ image ? Google Erth or other... Estimation of the areas where pre or post EQ imagery did not allow mapping ( because of clouds or shadows) | We have adjusted this |
| 13 | L164 : You said above you did not separate different zones of the landslides. How did you choose where was the initiation point? Is it the highest point? Taking a single pixel as source or scar zone may bias your statistics. Why not considering a scar surface in the upper part of the polygon? | This due to the limitation of our landslide inventories. For the Gorkha earthquake triggered landslide inventory, Roback et al (2017) identified the scarp areas of the landslide separately. For the RTL inventory we didn't do this. For the susceptibility assessment, we extracted the point located in the highest part of the landslides, as indicative of the initiation conditions. |
| 14 | L166 : Line 151 you said you use ASTER GDEM ( 30m). Be consistent. There is absolutely no reason to use a 90m dem while SRTM 30m is available. For quantitative slope assessment it will make a difference and analysis should be re performed with the highest possible resolution. | Different DEMs, such as ASTER GDEM, and SRTM Digital Elevation Model with both 90 m and 30m spatial resolution were evaluated to use in this study. After careful analysis however, both ASTER GDEM and 30m SRTM contained many erroneous data points, which forced us to use the more general 90m resolution SRTM DEM in our previous work. During revising this paper, we got another dataset, ALOS PALSAR DEM with resolution of 12.5m, which cover the whole study area. So the high resolution DEM was employed in this paper at last. |

| | | See line 163-169. |
|---|---|---|
| 15 | L167 : Explain how you determine where the river network start, as this is not done by arc GIS. | Base on the DEM, the streams were obtained using GIS modeling tool in ArcGIS and ILWIS software, and the drainage density was calculated. |
| 16 | L172: you mean it is from Shakemap ? At which resolution ? In any case a few sentences on how shakemaps are derived and on what are their limitations ( no topographic amplification, no constraints on site effects within mountainous area, interpolation with heavy weight given to station measurements even in areas with very different setting ) is needed, together with a couple of references. I also think a map of the shaking in the Koshi, with landslides indicated, should be shown at least in supplement. | The Peak Ground Acceleration data for the Gorkha earthquake were obtained from USGS Shakemap, which was designed as a rapid response tool to portray the extent and variation of ground shaking throughout the affected region immediately following significant earthquakes (Wald et al., 1999). We include the map in Figure 6 |
| 17 | L183: Did you use distance to river (as suggested above) or not? What is relative relief, computed at which scale? Same drainage density? Distance to fault, which faults? I think a supplementary figure with the different (relevant) susceptibility factor would be useful. | According reviewer's suggestion, we added a figure (Figure 6) that shows all contributing and triggering factors.

See Line287. |
| 18 | L207-210: which method did you use to determine the Beta exponent and the threshold size ? Clauset et al. 2009 is the recommended approach (and they provide script to reproduce their analysis). Are the different estimates significatively different (i.e., what are the uncertainty on them)? ETL-All and the two RTL dataset have very close exponents. | Indeed we used the method by Clauset et al (2009) , based on a script developed by Tanyas et al. (2018). From our new analysis based on the new landslide inventory for Gorkha earthquake, we found that, the ETL-All and ETL-Koshi have similar Beta exponent with value of 3.22 and 2.85, and the RTL landslide inventories have lower value of 2.38 and 2.44. What interesting is that all of them have similar cut-off value which round 30,000 $m^2$ |
| 19 | P9 214: landslide size definition : is there a mistake or this | We have adjusted this. |

| | | |
|---|---|---|
| | classification is discontinuous ? small <1000 ; 1000< medium <10,000 ; <100,000 large... What about landslide between 10,000 and 100,000?? | See line 226-229. |
| 20 | L216: why 6000? you say it is based on FAD but without explanation... The sentence above is meaning less, which FAD analysis? Which field exp ?6000 i the power-law cutofffor ETL but is in the roll over of RTL.... Also a few sentence on the meaning of the roll-over (and its sensitivity to resolution censoring) and of the Beta exponent and how it may be linked to physical properties is needed ! Cf Pelletier 1997, Stark and Hovius 2001, Stark and Guzzetti 2009, Frattini and Crosta 2013, | We have adjusted this. Based on the use of the ETL inventory (Roback et al, 2017) for the Koshi Basin we derived similar cut-off values of 30,000 $m^2$ for ETL and RTL.   Koshi River basin show similar cut-off value, which was around 30,000 $m^2$. So we defined the cut-off value as the threshold for large size landslide and small size landslide.

See line 211-229. |
| 21 | L224: For this initial correlation did you use ETL or only RTL ? If ETL was used what about PGA ? | We took PGA and precipitation factors as triggering factors, other factors we took as contributing factors. There are many groups during these factors. Here we only analysis some contributing factor to show the difference of different triggers and different sizes of landslides. |
| 22 | L229-231: I am not sure this comparative analysis in terms of altitude or other parameters make any sense : because the difference will not have any thing to do with EQ or Rain , just to the fact that one dataset (RIL) covers 10-20 times more area, with a vast area at low elevation. Instead the ETL are limited, because of the fault location, to a small zone with high elevation. I think all this analysis should be redone : ETL and RTL should be compared to the landscape within which they occur, so that it is not absolute elevation or | We agree that, the number of landslides in one landscape class can't show the correlation of landslide with the parameter, the density or frequency ratio could be better to show the impact of factor to landslides. Frequency Ratio was employed to show the impact of each factor groups on landsliding(Lee and Min, 2001; Razavizadeh et al. 2017). FR=(E∕F)/(M∕L) Where E is the area of landslide in the conditioning factor group, F is the area of landslide in the study area, M is the area of conditioning factor group, and L is the area of study area. Fig. 5 was redrawn and is now showing the Frequency Ratio for two combinations |

| | | |
|---|---|---|
| | slope or aspect that is analyzd but fraction of the lanscape (percentile of landscape elevation for exemple, or analysis of oversampling or undersampling of given slopes or aspect. Cf Meunier 2008, Barlow 2016 etc). Fig 5 should also be updated. | of contributing factors: elevation&slope and lithology&slope.

See section5.2. |
| 23 | L234: Is this based on the land cover maps ? Or is this from the imagery ? | This conclusion was drawn from image interpretation and field work. |
| 24 | L244: Missing word...to the?? direction ? | The word *South* was added |
| 25 | L264: gully density ? Or drainage density ? Be consistent ! | The word gully density was changed to drainage density. |
| 26 | L267-269: Could you comment on the values given for the different model ? It reaches 24 / 22 for ETL against 7 /6 for RTL. The methods sequence could include some more details to allow the author to have an intuition about the relative importance of different parameters | The coefficients for the contributing and triggering factors in the landslide susceptibility models show differences between triggers and different sizes of landslides. Curvature, altitude and slope gradient have a high impact on the susceptibility of RTL, while curvature, PGA, relative relief, and slope gradient have high impact on susceptibility of ETL. The size classes of RTL show larger differences in weight of curvature, relative relief and altitude. For ETL the difference between size classes are largest for factors of PGA, curvature, and relative relief.

See line 279-283. |
| 27 | L284: Obviously landslide susceptibility of ETL is giving only high suscptibility where you had data... As mentionned above you should also show the Shaking map ... | We add PGA map in the new figure, Fig 5. |
| 28 | L289 : EQ without effect on large landslides ? The argument that large landslides occur only close the fault may be true for very large landslides but seems unikely for | After the new analysis, we obtained a threshold of 30000m$^2$ for large size landslides. The characteristics and susceptibility zones show significant differences for small size and large size landslides. |

| | | |
|---|---|---|
| | landslides down to 6000m2 that is not
so large. | |
| 29 | L335 : I think this conclusion is erroneous, or at least not demonstrated by the authors. Because the ETL model includes PGA, and also because it is based on a much smaller part of the landscape, a subarea where landslide are located in a different environent copared to the zone affected by RTL. I think only by limiting the model development in an area where both RTL and ETL are widespread could the authors try to test this hypothesis.

*The conclusion that can be drawn is that the regions with very high and high suscepbility to ETL are not prone to RTL. This might change however, in the coming period, as the earthquake triggered landslides are bare and often the source of loose debris, that can be reactivated by extreme rainfall events.* | We fully agree with your statement and adjusted the text accordingly.

See line 371-436. |
| 30 | L340: You repeat this result that is completely obscure in the main text. There was no reason given to this threshold value | After reanalysis we are using a different threshold based on the cut-off points of the FAD's for both ETL and RTL, and explained this in section 5.1. |
| 31 | L342 : You never demonstrated the correlation in altitude and aspect was due to precipitation... The following sentences are interesting but a bit weak. The use of some | We add the average precipitation data during the monsoon season in Figure 6J.
And we also added text about this in the document in several section, including the discussion and conclusions. |

| | rainfall climatology ( as existing with TRMM for example, would be an actual demonstration). | |
|---|---|---|
| 32 | L349: Should be rephrased. The epicenter is extremely far from your study area and seismic waves propagates in all directions. Second part may refer to seismic directivity that relates to wave interference. I think a discussion in terms of the ground motion pattern is what you mean. (and it is difficult to discuss without showing the shaking in a figure...) | We add the PGA map in Figure 6h. |
| 33 | L365: The forcing extent are different within this catchment. You need to discuss it, and for that you need to show shaking and rainfall pattern, both essential information that are missing ! | See above |
| 34 | L368: "Some more detail information could be included in large scale research"
>> Like what ? why would it help? and why didn't you include it ? As of now this sentence does not bring anything to the reader. | Due to the new organizing of the manuscript, this sentence was deleted. |
| 35 | L376: "Whereas, the use of rainfall-triggered landslide maps can be of some use for predicting the occurrence of earthquake-triggered landslides, one should be careful, as the specific location of the earthquake plays a dominant role."
>> Not sure Whereas is the proper word. Anyway I do not think there is anything new for the community in a sentence like that | Due to the new organizing of the manuscript, this sentence was deleted. |

| 36 | Fig 5 : Why show Altitude vs other param? This display does show nicely the difference in altitude between datasets but not really with the other parameters. Further it is hard to interpret anything when the distribution of landscape parameters is not shown... I think the authors must show the distribution of landscape properties (as classically done in the literature) slope gradient, aspect, altitude; stratum etc in the study area in black and then the ones of landslide RTL/ ETL in red / yellow on top for comparison. | As mentioned earlier we have reanalyzed this and now show the Frequency Ratio for two combinations only : elevation&slope, and lithology&slope for both size groups and triggers, which is clearer.

See section 5.2 |
|---|---|---|
| 37 | Fig 6 : I suggest that you put all RTL in the left column and ETL in the right. It will make the figure less confusing and subplots easier to compare. | As reviewer's suggestion, we put all RTL in the left column and all the ETL in the right column, the figure is much clearer than before. |
| 38 | Fig 8 : large RTL are better predicted. Do you think this is physical or it may be a bias due to the higher difficulty to map small landslides? Also is there any ROC difference between RTL at any size and the 1992 or 2015 inventories ? | The large landslides are fewer, but seem to be related to a more defined set of combinations of contributing and triggering factors. This makes that the AUC's are higher. We didn't check the difference between the two RTL inventories separately. |
| 39 | Fig 9 : Comparison is not ideal : ETL susceptibility is likely driven by the fact landslides are limited to a very small subset of the Koshi. | According to the new inventories, the subset is not so very small: out 25,020 landslide, 14,127 were located in the Koshi river basin |
| | | |
| | **Technical comments** | |
| 1 | L85/86 : "are" missing between volume smaller/larger | "are" was added in this sentence. |
| 2 | L97 -> hazard and risk assessment (i.e. remove 1 assessment) | The extra assessment was deleted. |
| 3 | L214: From the Biblio it should be Tong et al. 2013. Given | We added in the text that this is considered a main reference in China for defining |

| | that this is a book in Chinese I doubt that this references will be accessible by much reader... and not sure it is essential . | the size thresholds |
|---|---|---|
| 4 | L 385 : Weather > Whether | The word weather was changed into whether. |
| 5 | Fig 7 caption : "Statistics" ; susceptiblity x2 > missing "i" | The word susceptibility was revised. |

**Response to RC 2**

| Comments | | Response |
|---|---|---|
| **Major comments** | | |
| 1 | Line 53-56:

Landslides were mapped from eight satellite images covering a period between 1996 and 2001 and concluded that the density of rainfall-triggered landslides increased significantly after the earthquake, and the places where landslides occurred changed, and concluded that different triggers produced significantly different patterns, with rainfall-triggered landslides occurring more near channels and earthquake-triggered ones close to ridges.

Long sentence. Rephrase | This reference was deleted due to the new structure of introduction.

See Line 58-73. |
| 2 | Line 85:

Missing reference at the end of the manuscript | The reference paper was added line 472:

Fan X Y, Qiao J P, Meng H, et al. (2012) Volumes and movement distances of earthquake and rainfall-induced catastrophic landslides. Rock & Soil Mechanics, 33(10):3051-3058. |
| 3 | Line 136 a and b missing | We added to the figure |
| 4 | Line 141:
Not clear chapter, highlighted sentences need to be reviewed and rephrased. Add resolution of used satellite images | This section was re-edited, see line 137-178 |
| 5 | Line 166 | For the susceptibility assessment, we extracted the point located in the highest part of the landslides, as indicative of the initiation conditions. |

| | | | |
|---|---|---|---|
| | how does the low resolution of the used data affects the reliability of the study? | Different DEMs, such as ASTER GDEM, SRTM Digital Elevation Model with both 90 m and 30m spatial resolution, as well as ALOS PALSAR DEM were evaluated to use in this study. After careful analysis however, both ASTER GDEM and 30m SRTM contained many erroneous data points, ALOS PALSAR DEM with highest resolution of 12.5m, was utilized in this study. ESRI ArcGIS software enabled the calculation of topographical factors including slope gradient, aspect, and curvature. Streams and gullies were obtained through DEM processing, and the drainage density was calculated.

See line 163-169 | |
| 6 | Line 175

"." Was missed | We added it. | |
| 7 | Line 185,186

Introduction to R and ROC | We added descriptions and references to R and ROC.

Fawcett T (2006); An introduction to ROC analysis. Pattern Recognition Letters 27:861–874 | |
| 8 | Line 207

Explain β | We explained β and added some references.
Size statistics of landslides are analyzed using frequency-area distribution curves of landslides (e.g., Malamud et al., 2004). There is a large literature arguing that frequency-area distribution of medium and large landslides has power-law distribution, which diverges from power-law towards smaller sizes (e.g., Hovius et al., 1997; 2000; Malamud et al., 2004). Given this argument, we can identify the divergence point of frequency-area distribution curve to determine a | |

| | | |
|---|---|---|
| | | site specific threshold values referring to the limit between medium and small landslides.

See line 211-215. |
| 9 | Line 218

For the value of 6000 | Base on FAD method, we analyzed the cutoff value, comparing the value with other's work, we get this value, but we changed the value to 30,000 according to our new analysis.
See line 216-229. |
| 10 | Line 244

ADD SW | Base on our new analysis, we change the description for this part. |
| 11 | Figure 6

Explain k2 and k1 | We have removed k1 and k2 from the figure. Figure 1 already shows the physiographic units. |
| 12 | Line 304 | We modified the description for this part:

The areal coverage of the landslide susceptibility classes was calculated for each susceptibility map (Fig. 9). Compared to RTL, the ETL susceptibility maps have a larger area with low susceptibility, due to fact that the Koshi River basin is far from the epicenter of Gorkha earthquake, thus the earthquake affected region is only part of the basin. The very high and high susceptible region for ETL is mostly concentrated in the western and southwestern parts of the basin, clearly reflecting the PGA pattern (Fig 6i). The RTL susceptibility also reflects the triggering factor (monsoonal rainfall), with the highest |

| | | susceptibility in the south of the basin. However, the higher rainfall peak in the Middle and High Himalaya region is less pronounced in the susceptibility maps, as well as in the inventory maps (Fig 3). The higher susceptibility classes for large ETL occupy more area than for small ETL, while the opposite can be observed for RTL.

See line 325-331 |
|---|---|---|
| 13 | Line 335

move it in the conclusion paragraph | We improved discussion and conclusion part dramatically.

See 371-436 |

**Response to SC 1**

Thanks for Dr. Scaringi's valuable comments to the paper at first. His comments were very useful to increase the quality of the paper.

(1) line 145 - I understand that the inventories were made through visual interpretation. It would be good if the authors specify this here rather than at line 150 (which refers only to the most recent images). Furthermore, it would be good to specify if and how the authors evaluated the mapping uncertainties due to low imagery resolution and visual interpretation, for instance in terms of shape and size mismatch and amalgamation, and their propagation to landslide statistics (e.g.frequency-area distributions, classification by controlling factors).

**Response:** Indeed, we agree with your comment, and modified the text. The landslide inventory pre-2015 was based on three data sets. The pre-2015 inventory map was generated using topographic maps, multi-temporal Google Earth Pro images and Landsat ETM/TM images. We were able to digitize landslide polygons from the available 1:50,000 scale topographic maps, which cover only the Nepalese part of the Koshi River basin. These maps were generated from aerial photographs acquired in 1992, and active landslides with a minimum size of 450 m$^2$ visible on these images were marked as separate units. A set of pre-2015 Landsat ETM/TM images were available for the entire study area, from which the post 1992 and pre-2015 landslides were mapped. Pre-2015 landslides were also mapped from historical images using Google Earth Pro Historical Imagery Viewer which contains images from 1984 onwards. Although the oldest images are Landsat images, the more recent ones have much higher resolution, although not covering the whole study area in equal level of detail. By comparing the different images for the period between 1992 and 2015 we were able to recognize most of the landslides. We carried out field verification for a number of samples and could conclude that through the image interpretation we were able to map landslide with a minimum size of 50 m$^2$.  Images from Google Earth were downloaded and geo-referenced and landslides were mapped using visual image interpretation and screen digitizing. A total of 5,858 rainfall induced landslides were identified in the Koshi River basin.

(2) line 168 - Also here, it would be good to specify how the rather low spatial resolution of the GlobeLand30 (30x30 m) affects the classification especially of landslides with small area (as low as 50 sq.m).

**Response:** We agree with your statement and we have also modified this in the text: Given the rather low resolution of the input data, the relation with landslides as small as 50m$^2$ may not be optimal, especially also considering the rather long time period over which land cover changes have occurred in many areas. But given the regional scale of this analysis, the use of higher resolution data was unfortunately not a viable option.

(3)  line 176 - Here it would be nice to explain the 60%-40% choice (is it because of the sample size? is it arbitrary?) and to specify how the landslides are assigned to either set (e.g. randomly, but being sure that the size distribution and controlling factors classification are the same in both sets?).

**Response:** Thank you for your comment. It is a generally accepted method in literature to separate the landslide dataset into a training and validation set (e.g. Hussin et al. 2016; Reichenbach et al., 2018). We decided to select 60% of the landslide data as training data for the modeling, and 40% for the validation. Here is comment on this matter from an expert on ResearchGate: "A common practice is to split the data set into L and T as 2 : 1. There is no profound justification for this; neither there is it clear, whether different splits yield less precise results. The result of a split is an assessment of the quality of the prediction by the model. Such an assessment is subject to uncertainty because the split entails randomness. An ideal split is associated with very small variation of the results. By a split we balance the uncertainty associated with the model (large L is preferred for that) and with evaluation (large T is preferred)". See also the below, from Hussin et al., 2016.

| Citations | Size of study area | Pixel resolution | Nr. of landslide pixels | Model ratio landslide : non-landslide pixels | Performance or validation rates |
|---|---|---|---|---|---|
| Van Den Eeckhaut et al. (2006) | 200 km$^2$ | 10 m | Training: 93 pixels
Prediction: 23 pixels | 1:5 | AUC ROC 0.91 – 0.98 |
| Hjort and Marmion (2008) | 600 km$^2$ | 25 ha (500 m) | 200 or more pixels | 1:1 | Mean AUC ROC 0.90 |
| Blahut et al. (2010b) | 450 km$^2$ | 10 m | Training: 21923 pixels
Prediction: 21923 pixels | 1:206 | AUC SRC: 0.87
AUC PRC: 0.88 |

| | | | | | |
|---|---|---|---|---|---|
| Regmi et al. (2010) | 815 km$^2$ | 10 m | Training: 368 pixels Prediction: 369 pixels | 1:22147 | AUC SRC: 0.77 AUC PRC: 0.74 |
| Van Den Eeckhaut et al. (2010) | 1120 km$^2$ | 50 m | 64198 pixels | 1:1 | AUC ROC 0.90-0.92 |
| Piacentini et al. (2012) | 7500 km$^2$ | 20m | Training: 617 pixels Prediction: 185 pixels | 1:30389 | AUC SRC: 0.80 AUC PRC: 0.76 |
| Felic śimo et al. (2013) | 140 km$^2$ | 10 m | 340 pixels | 1:2 | Mean AUC ROC 0.76 – 0.78 |
| Heckmann et al. (2014) | 19 km$^2$ | 5 m | 81 pixels | 1:3.7 - 1:4.3 | Mean AUC ROC 0.83 |
| Petschko et al. (2014) | 15850 km$^2$ | 5 m | 50 to 12562 pixels | 1:1 | AUC ROC 0.76 – 0.84 |

(4) line 216 - Here you classify the landslides into small and large depending on "field experience" and on the basis of the frequency-area distributions. You choose

6000 m2 as your threshold which is more or less the cut-off value in the frequency-area distribution of the earthquake-triggered landslides but is much smaller than that of the rainfall-triggered landslides. However, the cut-off (or rollover point) may be affected by under sampling of small landslides, which you should be able to rule out explicitly. Also, what field experience means in this context remains unclear. So, this threshold area seems quite arbitrary. I would encourage the authors to introduce a physically-based justification for this choice, which you did in part already in the introduction. On the other hand, I would also suggest that you run your model multiple times with different thresholds, to show if there is an optimal (data-driven) threshold that can best differentiate the statistics of RTL and ETL in your study area. This threshold will certainly have a hidden physical meaning, which could be then discussed

**Response:** The landslide inventories in the Koshi River basin show similar cut-off values, around 30,000 $m^2$ for different triggers (rainfall and earthquake). Here we should take in mind, however, that the two rainfall-triggered landslide inventories are not event-based inventories (Guzzetti et al., 2012 ). The two inventories differ in the sense that the 1992 inventory is based on landslides that were large enough to be mapped on the topographic map, where as the inventory between 1992 and 2015 represents the landslides that could be mapped from multi-temporal images over a number of years. Although the two inventories differ substantially with respect to the number of small landslides, it is striking to see that the cut-off values, and β values are similar. It is very difficult to obtain a complete event-based landslide inventory for rainfall inducedlandslides in Nepal, as landslides are generally generated by a number of extreme rainfall events during the monsoon, which can not be separated, as the area is cloud covered through most of the period.   The size-frequency distributions for both ETL and RTL show very similar behaviour for landslides above the cut-off value of 30,000 $m^2$. Landslides are generally classified in terms of area and volume.   But landslide volume is very difficult to measure, as it requires high quality multi-temporal Digital Elevation Models, and knowledge on slip surfaces (Jongmans and Garambois, 2007). In practice , landslide classification is mostly based on area, and in China the Tong et al. (2013) proposed a classification with landslides with an area smaller than 10,000 $m^2$ as small, those with an area between 10,000 $m^2$ and 100,000 $m^2$ as medium, and those with larger sizes than 100,000 $m^2$ as large size landslides. Based on the results of the FAD analysis, that resulted in similar cut-off values for the RTL and ETL and similar β values, we subdivided them into two size-groups, with 30,000 $m^2$ as threshold value (Table 1). The results will therefore be more reliable for the class above the threshold of 30,000 $m^2$ , where under sampling is not an issue, then for the small landslide class, which have different rollover points, and completeness levels.

**References:**

Guzzetti, F., Mondini, A.C., Cardinali, M., Fiorucci, F., Santangelo, M. , Chang, K.T. (2012) Landslide inventory maps: New tools for an old problem. Earth-Science Reviews 112:42–66

Hussin HY, Zumpano V, Reichenbach P, Sterlacchini S, Micu M, van Westen CJ and Balteanu D (2016) Different landslide sampling strategies in a grid - based bi - variate statistical susceptibility model. Geomorphology, 253: 508-523

Jongmans D, Garambois S (2007) Geophysical investigation of landslides: a review. Bulletin Société Géologique de France 178(2):101–112

Reichenbach P, Rossi M, Malamud BD, Mihir M and Guzzetti F (2018) A review of statistically-based landslide susceptibility models. Earth-Science Reviews, 180: 60-91

Tong L.Q., Qi L.S., An G.Y. et al. (2013) Large Scale Geological Hazards investigation by Remote Sensing Technology[M]. Science Press.

---

## Referee Report (RR1)

**Review of: "How size and trigger matter: analyzing rainfall-and earthquake-triggered landslide inventories 1 and their causal relation in the Koshi River basin, Central Himalaya" for NHESS.**

**Summary**

The paper addresses an interesting problem – the differing controls on earthquake versus rainfall triggered landslides. It provides a novel contribution by taking advantage of the co-location of many rainfall and earthquake triggered landslides in the Khosi catchment. In this respect the study is extremely exciting, offering the possibility to examine the influence of the different triggers while controlling for landscape properties. Given this, I think considerably more could have been made of the results both by examining the parameters used in the logistic regression in more detail, and by examining spatial susceptibility patterns at a finer scale. Finally, the examination of how landslide susceptibility varies with size is an interesting idea but I have major reservations (detailed in MC4) about how robust these findings are at present.

**Major comments**

**MC1.** One of the major difficulties for this study is comparing an event inventory (associated with an earthquake and its aftershocks) with a long-term 'historical' inventory that combines landslides triggered by multiple storms. This is unavoidable and does not prevent the study from having value, but it does introduce comparability problems between the inventories and the susceptibility maps that they generate. These problems need to be treated much more explicitly in the paper.

**MC2.** More discussion (and perhaps more analysis) is needed on how and why the earthquake and rainfall triggered inventories differ (e.g. L404). This is picked up by R1 in Detailed7 but has not been fully addressed. The key question is whether the difference is simply a result of the different spatial distributions of triggering intensity or of different triggering processes. The earthquake inventory is limited to a fairly small part of the study area where shaking intensities in the Gorkha earthquake were high. The ETL susceptibility map would perform poorly for almost any other earthquake you could choose. Differences related to the different processes should be visible in the relative importance of different predictor variables. You don't really examine this in your results (other than one sentence L276-8) but I think this is important to do. You should then discuss the relative importance of different predictors and how these compare to findings from other rainfall and earthquake triggered landslide studies (e.g. proximity to ridge crests, the relative importance of slope and aspect etc). This is particularly important because these parameters are the best and perhaps only tool that you have to address your aim of understanding how susceptibility differs with different types of trigger. The susceptibility maps themselves are in my opinion too strongly influenced by the particular trigger patterns (especially in the earthquake case).

**MC3.** Landslide susceptibility features throughout the paper and is central to your conclusions. It would be useful to include: a definition of landslide susceptibility in the introduction (see also R1Detailed3, which is only partially addressed); and a paragraph in the results that helps the reader to interpret the landslide susceptibility classes. Are these purely relative classes or is there some absolute interpretation to the values? If so what is the basis for it, if not how should we compare RTL and ETL susceptibilities or those for different size classes? What are the landslide susceptibility maps showing susceptibility to? I think this is to landslide initiation, in which case the step to hazard (being hit by a landslide) involves an assumption that locations with higher susceptibility have higher hazard. This is likely to be true at some spatial scale of aggregation but may not be true at the finest ~12 m resolution of your input data.

**MC4.** I think you are conflating the effect of landslide size and the effect of sample size in your analysis. To properly test whether large landslide predictors are particularly dissimilar from small landslide predictors you should also examine the extent to which small landslide predictors differ from one another. You could do this by taking multiple (m) subsamples of the small landslide dataset (with n=355 landslides per sample i.e. equal to the number of large landslides), then repeat your susceptibility analysis m times using each subsample. The resultant distribution of landslide predictors (i.e. the distribution of m values for each logistic regression parameter) could be compared to the large landslide predictors (i.e. the value of each logistic regression parameter for large landslides) using standard statistical testing. Without doing this I don't think you can make your key concluding claim that: "the resulting susceptibility patterns are quite different, and it is therefore questionable whether landslide susceptibility maps that are generated for all landslide size would be able to accurately predict the large landslides." (L424).

**Detailed comments**

L40, it would be useful if you could define 'size', for some it may be synonymous with 'volume'

L54, these explanations of differing size-frequency distributions are methodological but there are have also been attempts at mechanistic explanations. R1 points this out in Detailed Comment 20 but this part of the comment has not been addressed.

L144, "from which the post 1992 and pre-2015 144 landslides" there is a word missing at the end of this sentence.

L151, what is the resolution of the inventory (i.e. minimum mapable landslide size) and what are the planimetric errors for the boundaries of landslide polygons?

L159, as R1 points out (R1Detailed13), taking a single pixel as source or scar zone may bias your statistics. Why not considering a scar surface in the upper part of the polygon (as others have)?

L164, what algorithms did you use to generate these different metrics? Add references.

L164, "Streams and gullies were obtained through DEM processing" add details of the processing including parameter choices e.g. channel initiation threshold. R1 made this point in their previous review (R1Detailed15) but it has not been addressed.

L189, as R1 previously asked: "What is relative relief, computed at which scale? Same drainage density? Distance to fault, which faults?" (R1Detailed17), these questions have not been addressed.

L178, how did you allocate landslides to training or validation sets? This point was also raised by Dr Scaringi in comments on a previous draft but has not been fully addressed. Are these random samples? What additional constraints are you putting on the sampling (e.g. retain the original size distribution)? Did you test the sensitivity of your results to the particular random sample? E.g. by repeating the analysis for a second realisation of the training-validation partition? Given the problems I suggest either demonstrating that several realisations of your cross-validation give the same result or using a more robust approach (e.g. k-fold cross-validation or similar).

L183: How and why were these size groups chosen and what were the boundaries? Why did you choose only two groups? I don't necessarily disagree with the choice but I think that it needs to be explained. You go on to do this in section 5.1 so you just need to point the reader to that section here.

L184: a little more information is needed, what types of bivariate analysis?

L190, I don't think this sentence is correct: "For the susceptibility modeling of ETL, precipitation during monsoon(x10) was instead of peak ground acceleration (x10)."

L244, Frequency ratio estimates can be very sensitive to sample size within each conditioning factor group. I suggest looking at the approach of Rault et al. (2018) as a way to identify bins where sample size restricts your confidence in the frequency ratio estimate (details are in Supp info).

L245, how did you choose the number (and boundaries) of groups associated with each conditioning factor?

L245, I don't understand how you get: E, the area of landslides in the conditioning factor group and F, the area of landslides if your landslide dataset has been transformed into a set of points for the highest part of each landslide (L159).

L253, the comment re L244 is particularly true for multi-dimensional frequency ratio analysis so I would suggest the same approach here.

L268, what is the resolution of the susceptibility maps that you generate and how do you handle differences in the resolution of your predictor variables?

L286, fig 6, what data did you use to define the fault traces that you use in your distance to fault map fig6f? What choices did you make about the types of fault (e.g. age, slip history, size) that should be included?

L294, table 2, it would be useful to include the description of the predictor (e.g. slope) in the column headings.

L363, I don't think it is a good idea to combine discussion and conclusions. A summary of your main findings would considerably improve the paper.

L406, I don't think this is strong enough, 'may not be likely' the probability of the next earthquake producing the same shaking pattern is tiny!

L409, "However, using PGA values based on probabilistic seismic hazard assessment might result is relatively poor statistical correlations with event-based inventories." This needs explaining and or supporting with a reference.

References

Claire Rault, Alexandra Robert, Odin Marc, Niels Hovius, and Patrick Meunier, Seismic and geologic controls on spatial clustering of landslides in three large earthquakes, 2018, https://www.earth-surf-dynam-discuss.net/esurf-2018-82/

---

## Author Response (AR2)

Firstly, we want to thank the reviewers' careful reviewing and good suggestions to the manuscript. The reviewers have wealth of experience in landslide inventory and analysis; their professional comments were very helpful to improve the quality of this paper. We would like to express our sincere thanks to them. We tried our best to reply these comments one by one.

| Comments | Answer |
|---|---|
| **Main Comments** | |
| 1 The authors better explain the methods and data used to obtain their inventory of RTL. This is an improvement but the author fail to acknowledge the limit of their inventories due to resolution and revegetation issue. This must be stated, with a reference to the recent manuscript by Marc et al., 2019, where in a fraction of the study area of the authors (the Bhothe Koshi valley) hundreds of landslides were mapped between 2010 and 2014, while the authors catalogue (1992-2015) contain only a few in this zone (Fig 3). | We check our landslide inventory within Bhote Koshi valley; 300 hundred of landslides were interpreted. .During our interpretation, we also found that, the vegetation grows very fast in this area, many landslides regenerated in short time after they occurred, especially for small size of landslides. Also our high resolution images for landslide inventory were not obtained every year during 1992 to 2015. These reasons may cause differences between different inventories. In Fig.3 we used the landslide polygons to draw the map, because the outlines of landslides were very thin, and the scale of the maps was very small, so the landslides were not clearly visible. We adjust the landslide distribution map of Fig. 3. |
| | In our manuscript we added the limitation of RTL inventory in Line 156-158. |
| | *Main limitations affecting the landslide inventory are ought to a) revegetation on the areas of the landslides that occurred in 1992 and 2015 that impedes their detection on remote sensing images and b) lack of multi-temporal high resolution images in the region (Marc et al., 2019).* |
| I think the author present a somehow biased discussion on the two results of their susceptibility maps : I agree that RTL and ETL susceptibility maps are driven by different factor. In large part because PGA is very important for ETL, not RTL. But RTL susceptibility map is in the end almost not controlled by monsoon long term precipitation, that should be a little discussed by the authors. As a result this map does not really differentiate, in its prediction; RTL or ETL. So I think the authors need to state and discuss more rigorously their results: RTL susceptibility does not depends much on monsoon, thus either meteorology is less important than in situ parameters either other meteorological parameters should be used (e.g., monsooon | We would like to thank for the reviewer's deep analysis on this issue and suggestions. In the RTL susceptibility assessment, mean monsoon precipitation was taken as the rainfall triggering factor. This factor can only be used to indicate a general tendency for the landslide distribution at regional scale, Instead, most commonly in other studies, as indicator, the daily rainfall on the date of the event and the antecedent rainfall are correlate better with the landslide occurrence In the discussion part, we tried to discuss this issue base on reviewer's |

| | |
|---|---|
| variability, cf Deal et al., 2017). This should be explored in future studies. As a result, the RTL maps give a static susceptibility maps, that does not really discriminate landslide from a trigger or another. | suggestion and some references in Line 411-418:

*It should be clarified that although, commonly, the daily and the antecedent rainfall are used to describe the rain effect on the landslide occurrence, in this work, what is used is the mean precipitation during the monsoon season. The use of this value is chosen to provide, at regional scale, a general tendency of the landslide distribution. In the RTL susceptibility assessment model, the weight of the precipitation factor is low, which means this factor was not strongly correlated with the landslide susceptibility. As a suggestion, the use of the daily rainfall instead of the mean precipitation during the monsoon is preferred, in order to take into consideration its variability, as the use of the short-term rainfall variability to study the long term historical landslide inventory and susceptibility assessment may more reasonable (Deal et al. 2017).* |

Line by Line Comments

| | | |
|---|---|---|
| 1 | L72 : Please rephrase : They are mostly mapping landslide after an earthquake… But not Marc et al., 2015, that mapped landslide for several years before earthquakes in Taiwan, Papua New Guinea and to a lesser extent Japan.
Also do not cite Marc et al., 2015 for the Wenchuan earthquake that is no considered in this study.
I think Marc et al., 2019 is very relevant to this part of the introduction as it also focussed on RTL before the earthquake and in zones not affected by the earthquake. | We modified this sentence, and corrected the references. A new phrase was added in Line 72-74 as follows:

*There are fewer studies, carried out on multi-temporal RTL inventories in Taiwan, Papua New Guinea and Japan, which focus on the comparison of the RTL considering or not earthquake effects (Marc et al. 2015).* |
| 2 | L75-76 : Again you misuse Marc et al., 2015. The post earthquake RTL that are reactivation of coseismic landslides are very limited. There does not seem to be a clear correlation with coseismic pattern. The same is observed in Marc et al., 2019 where only 20-30% of RTL caused in 2015, just after the EQ, are spatially connected to ETL. | We modified this part in Line 75-80 as follow:

*The problem with the studies indicated above is that the rainfall-triggered landslides that occur shortly after a major earthquake are generally following the same spatial patterns, due to the availability of large volumes of landslide materials of the co-seismic landslides (Hovius et al., 2011; Tang et al., 2016; Fan et al., 2018a). However, other studies argue that there is not a clear correlation of rainfall-triggered landslides with the co-seismic pattern, as only the 20-30%* |

| | | |
|---|---|---|
| | | *of the RTL that occurred just after an earthquake, are spatially related to the ETL. The post-earthquake RTL that correspond to the reactivation of the co-seismic landslides are very limited (Marc et al. 2019).* |
| 3 | L95: I think your two questions can be more or less summarized in one : Because if they have the same control, the susceptibility of ETL and RTL should be the same, and one can be used for predicting the other. Opposite is expected if they are controlled by different factor you will not be able to use on to predict the others. | This part was rephrased in line 98-101:

 *The question that is addressed is whether different landslide sizes are controlled by different sets of contributing factors. Furthermore, it will be investigated whether it is possible to utilize inventories of earthquake-triggered landslides (ETL) as inputs for analyzing the susceptibility of rainfall-triggered landslides (RTL) and vice versa.* |
| 4 | L121 : Maybe mentioning Marc et al., 2019 here would be a good addition, as it gives the magnitude of annual landsliding in different High Himalayan valleys | Thanks for the suggestion, we added the reference in Line 127. |
| 5 | L141-145 and Fig 3:
RTL 1992 – 2015 : I think that you should mention here that the RTL landslide dataset are undersampling the actual amount of landslide during the 1992-2015 period:
Because of revegetation is rather rapid on landslide scar and possibly because of resolution. An example of that : Marc et al., 2019, mapped between 2010-2014 ~ 350 landslides in a 25x25km² of the Bhote Koshi valley, part of your study area.
Most, if not all of these landlides are not in you 1992-2015 mapping, while your imagery was just a few years after (so resolution or image quality is also likely at play).
If these zones had sustained landsliding at these rates during the last 20 years thousands of landslides are missing in this valley, and likely as much in the other valleys.

It is not necessary a problem for your study, but it is a bias that should be acknowledged, so that readers do not think it is a comprehensive representation of the landslide rate and location. (E.g., the sentence L148 is inaccurate and needs to be removed: "the different images for the period between 1992 and 2015 we were able to recognize most of the landslides") | We would like to thank the reviewer for this comment. We agree with the reviewer's suggestion.
During our inventory we also found that, the revegetation in this is area is very fast. Given the limitations in the resolution of the remote sensing images quality and obtained period, a complete landslide inventory in the whole area was not feasible. However the images were used to compile as much as possible the landslide database, to use it as a sample for this work.
Follow the reviewer's suggestion; we added some more phrases to the discussion part, concerning the limitations of the RTL inventory in Line 151. |
| 6 | L 211 : Ok but the size distribution also depends on how you define landslide area : For example Malamud 2004, explicitly state they remove all debris flow with long aspect ratios. Marc et al., 2019, found a power-law starting at 1000-2000 m3, when considering only | It is very difficult to separate the scar area from the runout area in this study. This was not possible for, the landslide inventory of 1992, which was digitized on the original topographic |

| | | |
|---|---|---|
| | landslide scar area (retrieved applying a correction on landslide runout, cf Marc et al., 2018).

>> So maybe mention these separation are relative and maybe improved by removing the area due to runout in individual and stuying landslide scar area only (Marc 2018, 2019) | maps, as high resolution images during this period were unavailable. Then, at regional scale, the differentiation of the boundaries betweenthe scar area and the landslide run-out could not be made,neither due to lack of high resolution images before the sliding., Morerver bias related to the experience of technical people who created the landslide inventory would strongly affect the results.

Because of these reasons, for this analysis we decided to use the boundary of the whole landslide without further differentiation between scar and run out. |
| 7 | L337 and Table 2: I am surprised by the statement the RTL reflects monsoonal control: In Table 2 you find a very small control of monsoon (x10 ~1) much smaller than the other types of control such as elevation (x1=7), slope (x2=6) or curvature (x3=-10). So the pattern of RTL | The rainfall factor used in the RTL susceptibility assessment was the mean precipitation during the monsoon season. But for the occurrence of landslides, the rainfall intensity is better correlated with the landslide occurrence. |
| 8 | L366 : Also limited because they are not comprehensive : See earlier comment on resolution and revegetation. | We add the limitation on RTL in this part in Line 377-379.

*Another limitation for this landslide inventory was related to the temporal and spatial resolution of the satellite images, as well as the revegetation the impedes the landslide detection for a complete historic landslide inventory* |
| 9 | L368 : Yes event trigger is a challenge. Note that a first database of RTL event inventories was published recently in Marc et al., 2018. | One sentence was added in line 379-381:

*There has been an increasing number of researchers working on the development of event-based landslide inventories and databases (Marc et al., 2018), which may be used to supply more samples for the comparison between RTL and ETL.* |
| 10 | L386 : "consensus" | This word in the manuscript was corrected. |
| 11 | L387-389 : Not only topographic, also mechanical properties (as underlined in Frattini and Crosta 2013 or Stark and Guzzetti 2009). Although the methods are somewhat different you could also mention that Marc et al., 2019 found similar Beta values between ETL and RTL, and also relatively similar to your (2.45-2.55).
The cutoff value is much smaller because a correction to remove runout was applied. | New sentence was added in line 400-404 as follow:
*Our findings regarding similar cutoff values obtained from different inventories created for the same area are also supporting this argument. This conclusion is also supported by Marc et al., 2019, who found similar Beta values between* |

| | | |
|---|---|---|
| | This ask the question whether a landslide with a long runout is a large landslide ? | *ETL and RTL, but also a cutoff value which is much smaller, as the result of a correction to remove the runout areas from the landslide boundaries.*

For the question: This ask the question whether a landslide with a long runout is a large landslide? Landslides with long runout present certain peculiarities as the initial part of the landslide (Source area) is respectively very small, but further material sources exist within its runout area. If we only use the initial source area to define the size of landslide, the landslide size will not be representative. |
| 12 | L390 : " precipitation in the Monsoon for RTL, and PGA distribution for ETL) have major influence on the distribution of landslides and susceptibility zones "
You cannot state that ! This is true for earthquake but NOT for monsoon, see my comment on Table 2 where the effect of monsoon is very small compared to topography, slope, curvature etc.
This is also demonstrated by Fig 10 : ETL susceptibility separate quite well ETL in high susceptibility and RTL in low susceptibility. In contrast, RTL susceptibility does not really distinguish RTL and ETL. This is consistent with the fact that RTL susceptibility is mainly driven by static factor and thus relates to both trigger types. | We would like to delete the phrase. |
| 13 | L402: Ok ETL map is specific, but your statement about RTL are not correct : You state that the RTL predict modestly the ETL… But it predict the small ETL almost as well as the small and large RTL. So you cannot say that RTL has much specificity. | We would like to delete the phrase. |
| 14 | L414 : "cocoseismic" > coseismic | Corrected. |
| 15 | L415 : No Marc et al., 2015 did not say that post-earthquake RTL was following coseismic landslides, see other comments. | Considering the controversy between different studies, we modified this part. |
| 16 | Fig 10 : I think some of the values in % are wrong (relative to curve positions...) | We would like to thank the for reviewer for this careful review. We modified the figure and corrected the position of the values. |
| 17 | References used and not in the manuscript :

Deal, E., Favre, A. C., & Braun, J. (2017). Rainfall variability in the H imalayan orogen and its relevance to erosion processes. Water Resources Research, 53(5), 4004-4021.
Marc, O., Behling, R., Andermann, C., Turowski, J. M., Illien, L., | We added these references in the manuscript. |

Roessner, S., and Hovius, N.: Long-term erosion of the Nepal Himalayas by bedrock landsliding: the role of monsoons, earthquakes and giant landslides, Earth Surf. Dynam., 7, 107-128, https://doi.org/10.5194/esurf-7-107-2019, 2019.

Marc, O., Stumpf, A., Malet, J.-P., Gosset, M., Uchida, T., and Chiang, S.-H.: Initial insights from a global database of rainfall-induced landslide inventories: the weak influence of slope and strong influence of total storm rainfall, Earth Surf. Dynam., 6, 903-922, https://doi.org/10.5194/esurf-6-903-2018, 2018.

Stark, C. P. and Guzzetti, F.: Landslide rupture and the probability distribution of mobilized debris volumes, J. Geophys. Res.-Earth, 114, F00A02, https://doi.org/10.1029/2008JF001008, 2009.

---

## Author Response (AR3)

**Response to Reviewer**

| | Comments | Response |
|---|---|---|
| 1 | L23 : "The results show that the frequency -area distributions of rainfall-and earthquake–triggered landslides varied considerably, with the former one having a larger frequency of small landslides "

The current text in the result section (p10, 11) and discussion rather reports similar FAD : similar Beta, similar cutoff, and the small extra frequency of ETL at small size may not be reliable given it is well before the cutoff.
So I think you should rephrase to something like : "the frequency -area distributions of rainfall-and earthquake–triggered landslides have similar cutoff and power-law exponent, although the ETL might have a larger frequency of smaller one." | We accept the suggestion and rephrase as *The frequency-area distributions of rainfall- and earthquake–triggered landslides have similar cutoff value and power-law exponent, although the ETL might have a larger frequency of smaller one.* |
| 2 | L31 while susceptibility maps for different size of earthquake-triggered landslides were similar.
This last part of the last sentence does not seem to match the Figure 8/10 and the discussion and conclusion. I suggest removing it. | The sentence of *while susceptibility maps for different size of earthquake-triggered landslides were similar* was deleted. |
| 3 | L72-78 : I appreciate the effort to clarify the debate and statement of the different studies on post-earthquake landsliding.
However I think you should introduce Marc et al., 2019 in the study summary (i.e., L 73 : "Fewer studies … in Japan and Central Nepal, which focus on … Marc et al., 2015, 2019 ") | The sentence was rephrased as *Fewer studies carried out on multi-temporal RTL inventories in Taiwan, Papua New Guinea, Japan and Central Nepal before earthquake, which supplied good comparison study for RTL under the effect and without the effect of earthquakes (Marc et al. 2015, 2019)* |
| 4 | Further L 78-79 the sentence can be combined (remove the fact it is now not clear which studies give this value of 20-30% of spatial connection).
So I suggest to rewrite : However, other studies argue that there is not a clear correlation of rainfall-triggered landslides with the co-seismic pattern, as only the 20- 30% of the RTL that occurred just after an earthquake, are spatially related to ETL, suggesting limited re-activation of ETL by RTL (Marc et al., 2015, 2019).
With this edit the next sentence (L80) become redundant and can be deleted. | According to reviewer's suggestion, this sentence was rewritten as *However, other studies argue that there is not a clear correlation of rainfall-triggered landslides with the co-seismic pattern, as only the 20-30% of the RTL that occurred just after an earthquake, are spatially related to ETL, suggesting limited re-activation of ETL by RTL (Marc et al., 2015, 2019).* |
| 5 | L417 : " may be more reasonable" | This error was corrected in Line 424. |

| 6 | References : Stark and Guzzeti 2009 is added in the bibliography but not in the text. Maybe add it Line 399 | The reference was added in Line 403 of the manuscript. |
|---|---|---|

**Response to Editor**

| | Comments | Response |
|---|---|---|
| 1 | improve the legend sizes of the figures 6 and 8; | The legends in Fig. 6 and Fig. 8 were improved and clearer. |
| 2 | set up the conclusion in a separated chapter | *Conclusion was separated as chapter 9 in line 408.* |
| 3 | add, if it is possible, one figure taken in the field about the analyzed landslides. | Two photos on RTL and ETL in Koshi river basin were added as Fig. 2 in line 159.
The field work was supported by the "135" Program of IMHE (Grant No. SDS-135-1708), we added this program in the acknowledgments. |

[revised manuscript text omitted]